# Programming mechanics in knitted materials, stitch by stitch

Krishma Singal ®[1,5], Michael S. Dimitriyev ®[2,3,5], Sarah E. Gonzalez ®[1,5], A. Patrick Cachine[1], Sam Quinn ®[1] & Elisabetta A. Matsumoto ®[1,4] ✉

Knitting turns yarn, a 1D material, into a 2D fabric that is flexible, durable, and can be patterned to adopt a wide range of 3D geometries. Like other mechanical metamaterials, the elasticity of knitted fabrics is an emergent property of the local stitch topology and pattern that cannot solely be attributed to the yarn itself. Thus, knitting can be viewed as an additive manufacturing technique that allows for stitch-by-stitch programming of elastic properties and has applications in many fields ranging from soft robotics and wearable electronics to engineered tissue and architected materials. However, predicting these mechanical properties based on the stitch type remains elusive. Here we untangle the relationship between changes in stitch topology and emergent elasticity in several types of knitted fabrics. We combine experiment and simulation to construct a constitutive model for the nonlinear bulk response of these fabrics. This model serves as a basis for composite fabrics with bespoke mechanical properties, which crucially do not depend on the constituent yarn.

Knitting has long been regarded as an art that turns natural fibers into garments. Recently, engineers have begun to use knitting as an additive manufacturing technique to construct durable[1] textiles with bespoke mechanical properties and geometries[2] from 'yarns' made from a myriad of materials. Textiles research has traditionally been housed in both textile engineering and computer graphics; however, the growing interest of textiles as metamaterials[3] in other fields[4,5] creates the need for cross-disciplinary pollination. From that viewpoint, knitted textiles are mechanical metamaterials whose properties are imbued by the pattern of stitches, which exists irrespective of the choice of particular yarn. By choosing the appropriate stitches and their ordering, one can sculpt the local mechanical response of a textile using a yarn of their choice. Tunable compliance and tensile strength of knitted and braided structures made from bio-compatible yarns are used for medical bandages[6], surgical grafts[7,8], and mesh implants[9–12]. The mechanical properties of knitted textiles make them ideal for wearable electronics[13,14], soft actuators[15–21], as well as strain and pressure sensors[22–26] used in medical monitoring and therapeutics[27–29]. Likewise, knitted textiles can harvest energy from human movement[30–32] and even store energy as wearable supercapacitors[33,34]. By spatially varying the pattern of stitches, we can generate textiles with high or low stiffnesses (Fig. 1). With the aid of computerized knitting machines, we can program regions of variable stiffness into a larger textile. Unlike other composites, the entangled microstructure that gives rise to a knitted fabric's variable rigidity also holds it together along seamless interfaces. Continuously modifying the in-plane rigidity of a textile across a region can mitigate the damage often associated with large stresses at interfaces[35].

To facilitate the rational design of textiles, we need to understand the fundamental mechanics of knitted materials. Here, inspired by the design of hand-knit garments, we study how the mechanical behavior of weft knitted fabrics is encoded by the topology of their stitches as a first step towards creating a design tool for programmable textile metamaterials. The stitch pattern and mechanical properties of the

[1]School of Physics, Georgia Institute of Technology, Atlanta, GA 30332, USA. [2]Department of Polymer Science and Engineering, University of Massachusetts, Amherst, MA 01003, USA. [3]Department of Materials Science and Engineering, Texas A&M University, College Station, TX 77843, USA. [4]International Institute for Sustainability with Knotted Chiral Meta Matter (WPI-SKCM2), Hiroshima University, Higashihiroshima 739-8526, Japan. [5]These authors contributed equally: Krishma Singal, Michael S. Dimitriyev, Sarah E. Gonzalez. ✉e-mail: sabetta@gatech.edu

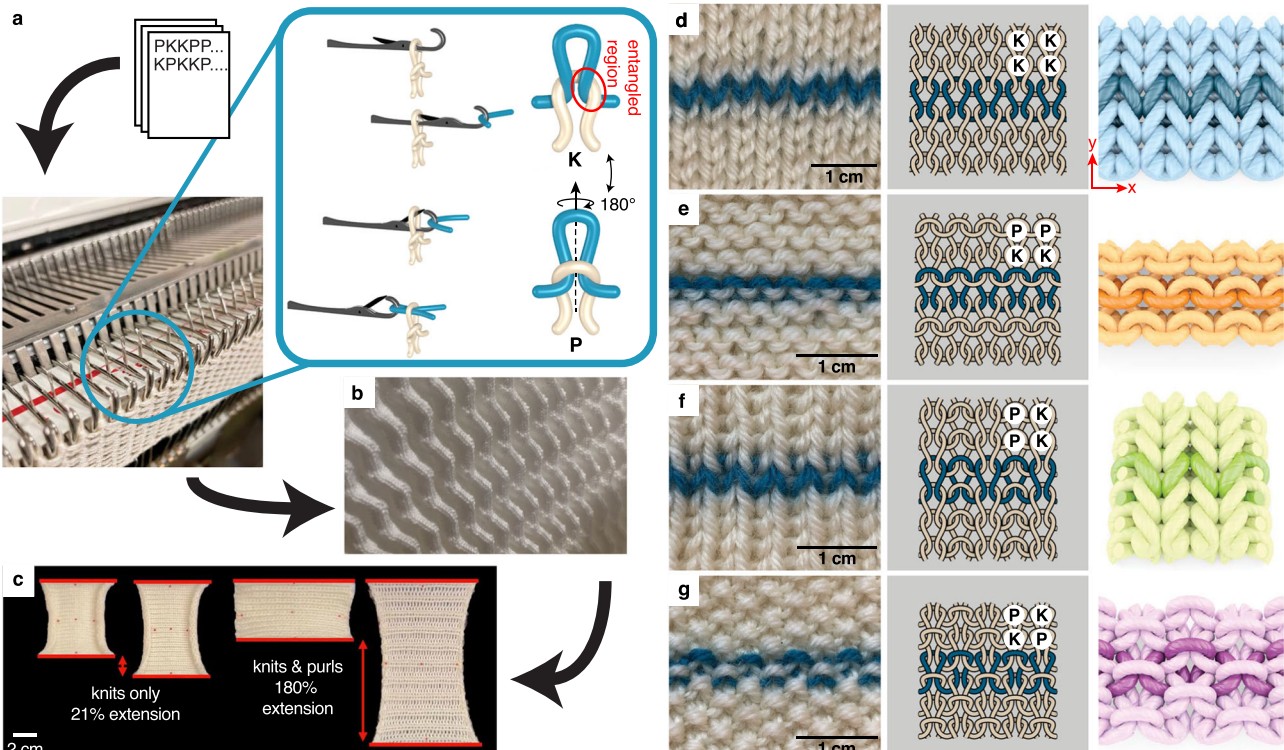

**Fig. 1 | Knitted materials have elastic responses that can be programmed by the pattern of Ks and Ps. a** A schematic of the knitting process where a knitting machine converts a code of **K**s and **P**s into a textile such as the Issey Miyake[67] sweater shown in (**b**). The knitting machine manipulates a bed of latch needles that pull new loops of yarn through existing loops to build the knitted fabric. Here, the second bed of the knitting machine (the ribber) has been removed for clarity. An entangled region of the stitch is identified by the red circle in the inset of (**a**). **c** Knitted fabrics with a mix of both **K**s and **P**s are markedly more extensible (under the same applied stress) than ones with only a single type of stitch. **d**–**g** Close up images (left), line diagrams (center), and simulation results (right) of four fabrics: (**d**) stockinette, (**e**) garter, (**f**) rib, and (**g**) seed.

constituent yarn are quasi-independent knobs we can fine tune. A consequence of our model is that knitting can be used to program mechanics at any lengthscale, from polymeric and colloidal assemblies[36] to light-weight tensile support in building construction[4].

The computer graphics community has made great strides in creating knit fabric simulations with visual fidelity[37–39], often with the goal of modeling entire sheets[40] of fabric and garments[41]. There is recent work on changing local fabric elasticity by changing the constituent yarn[41], but there has not yet been a systematic study of how changes in stitch topology affect the fabric elasticity[42] – even modeling stockinette (sometimes called jersey or plain-knit) fabric is quite complex[43–45].

In this work, our goal is to study knit fabrics from three different types of models: a minimal model of yarn-level simulation at the microscopic level, a constitutive model at the textile level, and our "Reduced-Symmetry" model at the intermediate level to unite these two points of view. Traditionally elastic response in knitted textiles is achieved by modifying the properties of the yarn often using blends of natural (wool and cotton) and synthetic fibers (polyester, nylon, or other plastics) which contribute to microplastic pollution[46]. To maximize extensibility, manufacturers reduce the amount of natural fibers used in the fabric and increase the amount of elastane and/or other elastomeric fibers. Our goal is to use stitch type as a way of modulating the bulk elasticity of fabrics made of inelastic yarn, irregardless of fiber composition, so that the desired elastic response of a textile can be achieved with natural and/or biodegradable fibers and without synthetic materials. Recent research has shown that a broad range of synthetic materials can degrade when in contact with skin secretions, which increases the potential for dermal absorption of compounds within those fibers[47].

## Results

### Topology and elasticity

Knitted textiles are composed of a rectangular lattice of slip knots. The two foundational stitches in knitting are the *knit stitch* (denoted **K**, also known as a front stitch) and the *purl stitch* (denoted **P**, also known as a back stitch). These two stitches form the bulk of a textile's structure, although many more complicated stitches exist[48]. The knit stitch is formed by passing a loop of yarn from the back to the front of the textile through an existing loop, while the purl stitch pulls the new loop from the front to the back. Therefore, knits and purls are fundamentally the same object, just related by a 180° rotation about the *y*-direction of the fabric (Fig. 1a). A schematic of the knitting process is shown in Fig. 1a,b. Combining **K**s and **P**s in different patterns generates textiles with markedly different linear elastic responses (Fig. 1c). Our goal is to untangle this relationship between stitch pattern and mechanical response using four common knitted fabrics: stockinette (Fig. 1d), garter (Fig. 1e, also known as links-links), rib (Fig. 1f), and seed (Fig. 1g).

The combination of entangled elastic segments and confinement makes knitted fabrics different from many mechanical metamaterials. The microstructure of a knitted fabric has *entangled regions* whose contact interactions dictate the stiffness and unconstrained regions that enable extensibility. Changing the ordering of yarn in an entangled region changes the topology of the fabric. Therefore, the topological method of knot theory is used to study textiles[48,49]. Previous studies have shown that the ordering of crossings within a knot can have a major impact on its strength[50], indicating a strong relationship between topology and mechanics.

We measured the elastic response of each of the four common knitted fabrics (Fig. 1d–g) in a series of uniaxial stretching

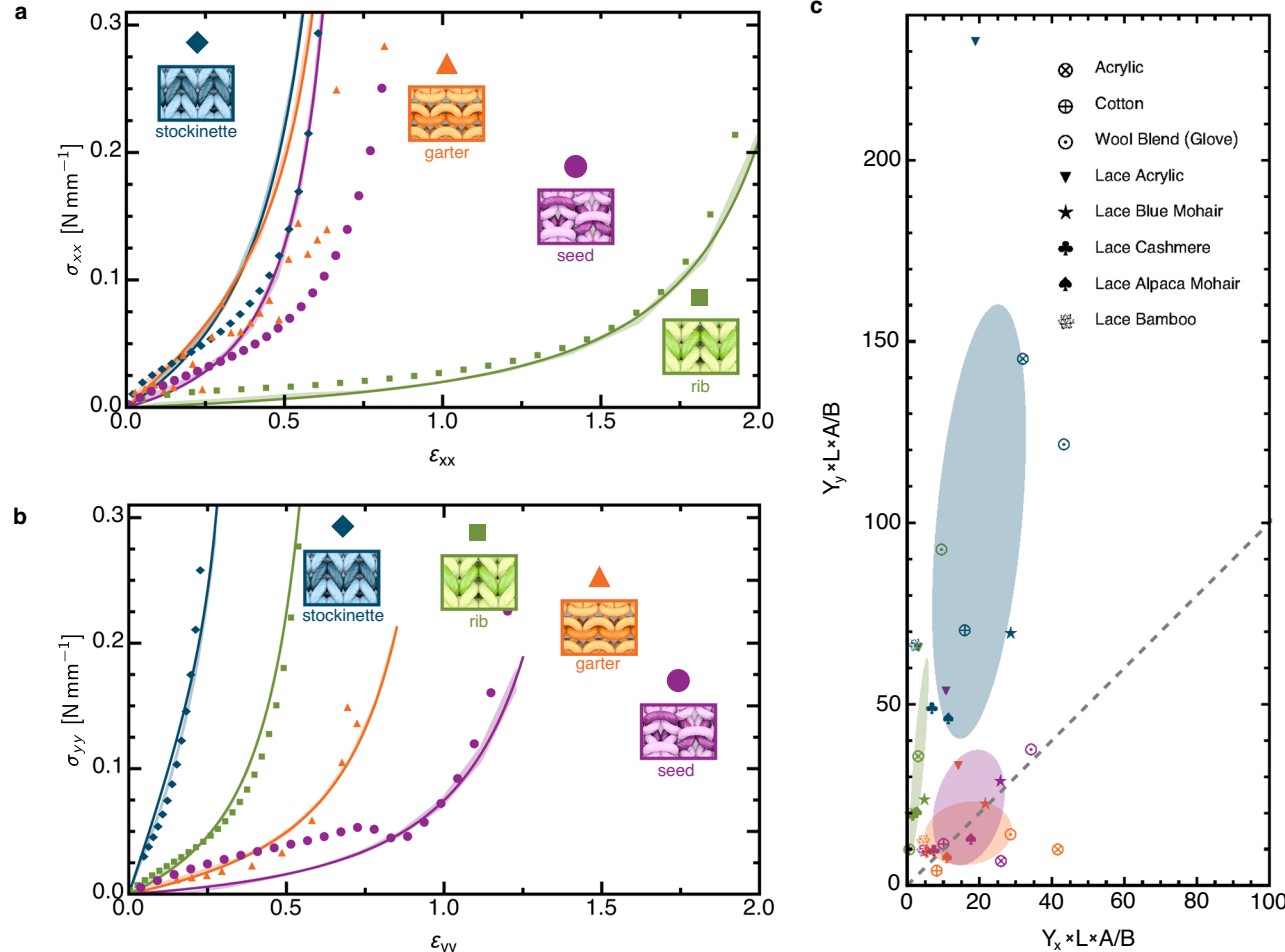

**Fig. 2 | Experimental and simulated results of uniaxial stretching.** The stress-versus-strain relations for the four fabrics made from the acrylic yarn in the (**a**) $x$- and (**b**) $y$-directions. All of the data for each type of fabric is displayed by a different color: stockinette in blue, garter in orange, rib in green, and seed in purple. The experimental data is shown in the translucent regions where the width of the region is one standard deviation of the four experiment runs. The simulation data is shown with solid symbols. The solid curves are fits to the constitutive relations. This is a system where the linear response for each fabric is significantly different despite only small differences in the stitch configuration, whereas the nonlinear parts are quite similar. Experiments applying force in the $x$-direction show the extreme extensibility of the rib pattern compared with the other three. Garter and seed dominate in the $y$-direction. Note, the experimental measurements for seed fabric differ from that of simulations due to a compression-related buckling instability in the computation, investigated in Supplementary Note 4 and Supplementary Fig. 8. **c** Normalized rigidity plot of all fabric samples, where $Y_i$ is the Young's modulus in the $i^{th}$ direction in N/mm (Supplementary Tables 10, 12, 14, 20), $L$ is the length of yarn per stitch in mm (Supplementary Tables 2, 3, 6), $A$ is the area of one stitch in mm² (Supplementary Tables 5, 6), and $B$ is the bending modulus in N mm² (Supplementary Table 1). The colored ellipses represent one standard deviation for each of the four types of fabric and are oriented along the principal axes. The gray dashed line represents an isotropic mechanical response. The same analysis was conducted on the un-normalized rigidities, shown in Supplementary Fig. 11. Source data are provided as a Source Data file.

experiments[51] and simulations (see Methods; Supplementary Fig. 1; Supplementary Fig. 3; Supplementary Tables 7, 9, 11; and Supplementary Notes 1–7). We fabricated and characterized samples made from two types of yarn, an acrylic yarn (Fig. 2) and a pearlized-cotton (Supplementary Fig. 2), which have different mechanical properties (see Methods). With the fabric under fixed uniaxial loading, we measured the bulk fabric deformation using computer vision[52,53] (see Methods and Supplementary Fig. 1). The maximal longitudinal components of the average stress $\sigma$ versus strain $\varepsilon$ measurements are shown in Fig. 2, where the $x$- and $y$-directions are along the rows and columns of the fabric (Fig. 1d).

Under small stresses, the responses of all the fabrics are linear, and the Young's moduli are given by the slopes of the stress-vs-strain curves (Fig. 2; Supplementary Tables 10, 12). Under high stresses, their responses become nonlinear, displaying strain-stiffening behavior as the yarn within the stitch becomes taut. Of the four fabrics, rib is by far the softest in the $x$-direction while stockinette is the stiffest (Fig. 2a).

Similarly, the garter and seed fabrics are softer in the $y$-direction (Fig. 2b). In Fig. 2c, we have plotted the normalized Young's modulus in the $x$-direction by the normalized Young's modulus in the $y$-direction for samples made from eight different types of yarn of varying sizes and constituent fibers. The clusters of data confirm that the relative anisotropy is fairly consistent across each type of fabric, regardless of the constituent fiber (Supplementary Fig. 11).

## Numerical model

Simulations help us unravel the effect that stitch topology and microstructure have on the macroscopic elasticity of the fabric. Stitch-level simulations (also known as loop modeling) have been of interest to a variety of fields, including textile engineering and metamaterials. Current simulations typically have at least one of three primary limiting factors: they do not consider compressible yarn[45,54,55], they only consider one type of fabric[20,45,54–56], or they only compare simulation to experimental results for visual fidelity and not mechanical

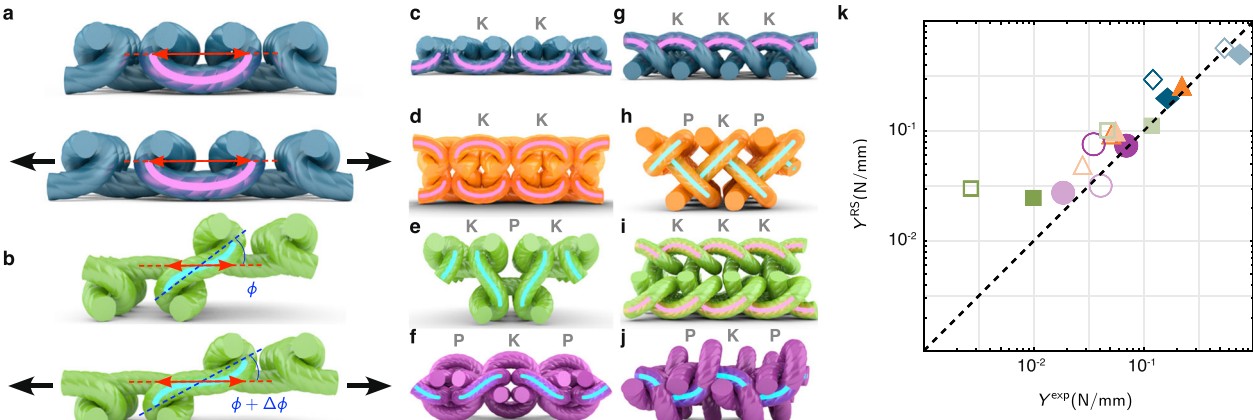

**Fig. 3 | Symmetry in the yarn segments between stitches.** Two similar stitches (**K**-**K** or **P**-**P**) are joined by a yarn segment with even symmetry, highlighted in pink (**a** top). Extensional deformations cause curvature deformations of the yarn segment (**a** bottom). Alternating stitches (**K**-**P**) are joined by a yarn segment with odd symmetry, highlighted in cyan (**b** top). These segments are able to rotate to accommodate extensional deformation (**b** bottom). Symmetries of stitches are shown in the *x*-direction (**c**–**f**) and the *y*-direction (**g**–**j**). **c**, **g** Stockinette fabric has only even connecting yarn segments in both *x*- (**c**) and *y*-directions (**g**). **d**, **h** Garter fabric has even connecting yarn segments in *x*-direction (**d**) and odd connecting yarn segments in the *y*-direction (**h**). **e**, **i** Rib fabric has odd connecting yarn segments in the *x*-direction (**e**) and even connecting yarn segments in the *y*-direction (**i**). **f**, **j** Since seed fabric is based on a checkerboard pattern, it only has odd connecting yarn segments. The renderings in (**a-j**) are repeated unit cells of sample stitch-level simulation outputs. A comparison of Young's moduli measured in experimental samples $Y^{exp}$ with those computed in the reduced-symmetry (RS) model $Y^{RS}$ (Supplementary Tables 17, 18) is shown in (**k**). Dark and light symbols indicate extensional rigidity in the *x*-direction and *y*-direction, respectively, filled symbols indicate acrylic yarn, and open symbols indicate cotton yarn. This demonstrates that our simple composite model has both qualitative and quantitative agreement with our experimental measurements. Source data are provided as a Source Data file.

response[37,57]. Our simulation method considers all three of these factors to investigate the role of stitch topology on the mechanical behavior of knit fabrics.

Yarn is an inherently hierarchical material with short staple fibers spun into indefinitely long yarn. To model the complex mechanics of yarn, we use yarn characterization experiments to measure the dominant energetic contributions: bending and compression (Supplementary Notes 2, 3; Supplementary Tables 1, 4). The torsional rigidity of a balanced, spun yarn is comparatively negligible, so our model allows the yarn to freely twist. Similarly, the extensional rigidity of the yarn is taken to be large so that the stretch of the yarn plays a minimal role in the fabric's ability to stretch. We simulate the yarn as a space curve $\mathbf{\gamma}(s)$ subject to a bending energy that is quadratic in the curvature, $E_{bend} = (B/2) \int_0^L ds \, |\partial_s \hat{\mathbf{t}}|^2$, where $s$ is the arclength parameter, $\hat{\mathbf{t}} \equiv \partial_s \mathbf{\gamma}$ is the unit tangent vector of the curve at each point, and the yarn parameters (the yarn length per stitch $L$, also known as loop length, and the bending modulus $B$) are measured experimentally (see Methods; Supplementary Note 2; Supplementary Fig. 4; Supplementary Fig. 9; and Supplementary Tables 1, 2, 3). To capture yarn-yarn interactions, we use an elastic core-shell model informed by experiments (see Methods, Supplementary Fig. 5, and Supplementary Note 3). This also prevents yarn segments from passing through one another. By implementing a minimal model in simulations, we can determine the key ingredients that contribute to the different mechanical behavior of different types of fabric so that our results can be efficiently utilized in the fields of mechanical metamaterials and extreme mechanics.

The periodic nature of knitted textiles enables us to reduce the system to a closed segment of yarn in a box with boundaries identified (Supplementary Fig. 6). We numerically minimize the total yarn energy, while varying simulation box dimensions (see Methods and Supplementary Note 4). Through our model, we effectively capture not only the geometry of knitted fabrics (Fig. 1d−g)[43,58,59] but the emergent elastic response as well (Fig. 2). The simulations reproduce the key features of the experiments: (i) the differences between the extensional rigidities of each fabric resulting from their unique topologies in the low-tension regime and (ii) the divergent strain-stiffening behavior corresponding to the maximum extensibility of each stitch in the high-tension regime. The simulations enable us to disentangle the ways in which contact energy and bending energy individually contribute to the local deformations of the yarn. In the low stress regime, bending energy is the dominant contributor to elastic response. In the high stress regime, compression energy shows a marked increase, as shown in Supplementary Fig. 7 and Supplementary Table 8.

## Microstructure and modulus

Knit stitches and purl stitches have fundamentally the same mechanical behavior. However, if we encode them – like binary bits – into a full textile, we see additional emergent behavior. In this way, we can view knit fabrics as a composite where each stitch has a fundamental elasticity and the yarn that connects each pair of stitches modifies the behavior based on its local symmetry. When two knit or two purl stitches are next to each other, they are joined by a connecting yarn segment which has even symmetry (Fig. 3a). When a knit stitch is joined to a purl stitch, however, the connecting yarn segment has odd symmetry (Fig. 3b). In the linear regime, the even and odd segments act as springs with different stiffnesses, as diagrammed in Supplementary Fig. 10c−f. We approximate the effective stiffness of the connecting yarn segments by taking its shape (from a fabric that has no forced applied to it) and calculate the work required to deform it infinitesimally (Supplementary Note 8; Supplementary Tables 15, 16). When we do this to linear order, we find that the symmetric region has a stiffness that approximately scales as $Y_{even} \sim 180B/[\lambda^3(1 - \delta_{even})]$, where $\lambda$ is the length of the segment (shown in Supplementary Fig. 10a, b) and $\delta_{even}$ is a geometry-dependent factor. The odd connecting yarn segment effectively acts as a moment arm where the two neighboring stitches apply a torque that causes it to rotate. To linear order, the stiffness is approximately $Y_{odd} \sim 12B/[\lambda^3(1 - \delta_{odd})]$. Therefore, odd connecting yarn segments can be of order ten times softer compared to even connecting yarn segments (see Supplementary Note 8). It is consequently harder to extend fabrics with identical neighboring stitches (**K**-**K** or **P**-**P**) than alternating neighboring stitches (**K**-**P**). This explains the relative stiffness of stockinette fabric,

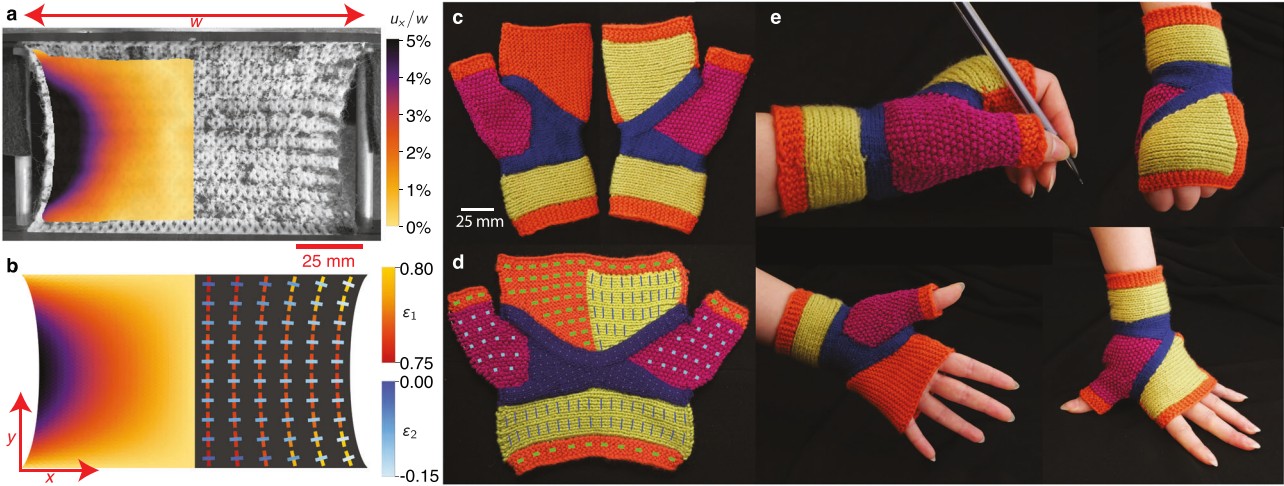

**Fig. 4 | The anisotropic and nonaffine global response of knitted textiles and an application of them. a, b** Large applied stresses result in nonaffine deformations to a knitted fabric. **a** The $x$-component of the displacement field ($u_x$), obtained from DIC measurements, is shown overlaid on an image of garter fabric. The color represents the magnitude of $u_x$, in units of fabric width $w$. **b** Finite element analysis (FEA) of our constitutive model reproduce the (left) displacement field seen in experiments and (right) the crosses show the principal directions and magnitudes of the local strain tensor. The values of local principal strains (scale bars in orange for $\varepsilon_1$ and blue for $\varepsilon_2$) show the degree of local extension and transverse compression. **c** Therapeutic glove prototype uses all four types of fabrics to generate anisotropic elastic response to motion of the hand. **d** The extensibility field of each type of fabric, shown as an overlay of rectangles, are oriented along the principal stiffness directions. The edge lengths are given by $1/Y_x$ and $1/Y_y$ respectively. **e** The stiffest stitch pattern, stockinette (blue), supports the wrist joint, while the isotropic seed (pink) grants mobility to the thumb. Highly anisotropic rib (green) and garter (orange) enable the wrist and fingers to flex along their easy direction.

consisting only of even connecting yarn segments (Fig. 3c, g), compared with seed fabric, consisting only of odd connecting yarn segments (Fig. 3f, j). Garter (Fig. 3d, h) and rib (Fig. 3e, i) fabrics each contain a mixture of segments but are much easier to stretch along the directions containing odd connecting yarn segments. A pair of similar stitches (**K-K** or **P-P**) joined in the $y$-direction are in general stiffer than a pair of equivalent stitches joined in the $x$-direction because the $y$-direction has two connecting yarn segments in parallel between every pair of stitches (Supplementary Fig. 10c–f).

Using the "rule of mixtures" from the theory of fiber composites[60], we build an effective elastic model for fabrics consisting of knit or purl stitches alternating with connecting yarn segments of the appropriate symmetry. We call this the Reduced Symmetry (RS) model. In the low stress regime, we are treating the fabrics as a composite of geometries, rather than a composite of materials. This allows for a direct estimate of the linear elastic rigidity using yarn geometry informed by simulations and bending modulus alone. To establish the dependence of the fabrics' anisotropic elastic response on stitch symmetry, we compare RS model estimates (using geometric parameters shown in Supplementary Tables 8, 9) of the Young's moduli to those measured in experiments while varying stitch pattern, direction of extension, and type of yarn (Fig. 3k) (Supplementary Note 8). Young's moduli estimated from our RS model closely agree with those measured in experiments, yet are systematically slightly stiffer.

In the high-tension limit, all yarn segments between neighboring entangled regions straighten along their mid-lengths and are forced to curve sharply as they enter the entangled regions due to contact confinement. This localization of curvature to entangled regions under increasing stress represents a transition from the low-stress, linear elasticity dictated by stitch topology, $\sigma_{low}(\varepsilon) \sim Y\varepsilon$, to high-stress, strain-stiffening elasticity, $\sigma_{high}(\varepsilon) \sim \beta(1-\alpha\varepsilon)^{-2}$, where $Y$ is a Young's modulus and $\beta$ and $\alpha$ are parameters characterizing the non-linear response. Each of these three parameters depend on the direction of extension. With this reasoning, we arrive at a stress-strain constitutive relationship $\sigma(\varepsilon) = \sigma_{low}(\varepsilon) + \sigma_{high}(\varepsilon)$ (Supplementary Note 6). Figure 2a, b and Supplementary Figs. 2, 13 show self-consistent fits of this model to our data. This model is able to describe all knitted fabrics made from inextensible spun fibers (Supplementary Tables 9, 11, 13, 19). This form

of constitutive model resembles the force-extension relationship for stiff, DNA-like polymers[61] as well as amorphous fiber networks[62].

## Applications

While our measurements and models capture the bulk constitutive properties of knitted fabric, the presence of boundaries can give rise to significant inhomogeneous response. The bulk constitutive model can nonetheless well-approximate the full deformation of a finite swatch of knitted fabric, as illustrated in Fig. 4a, b, where we compare the $x$-component of the displacement field of a sample of garter fabric stretched in the $y$-direction (measured using digital image correlation, DIC) with a finite element analysis (FEA) that applies our constitutive model to a two-dimensional sheet with more realistic boundary conditions without directly considering the local microstructure (Supplementary Note 9 and Supplementary Fig. 12). We used garter experiments to directly obtain fits to our constitutive model for use in the FEA, without homogenizing the yarn level simulations[40,63]. Notably, our constitutive model–derived from microscopic fabric properties–accurately captures the non-affine deformation of the fabric near its corners (where the principal stretch directions are no longer purely along the $x$- and $y$-axes) and reproduces the shape of the free boundary.

Emergent elasticity sets knitting apart from other additive manufacturing techniques, because merely dictating the local topology by interchanging knits and purls (not changing the constituent yarn) programs the fabric's local elastic response. We can take advantage of the local anisotropic response of each different type of fabric by combining them into a seamless garment, in this example a prototype for a therapeutic glove (Supplementary Note 10 and Supplementary Fig. 14). The goal of our prototype is to direct the stiff elastic response to support the wrist joint in cases of repetitive stress injury, while enabling natural motion for the rest of the hand (Fig. 4c–e). In Fig. 4d, the local extensibility field is represented with rectangles oriented along the principal directions with side lengths given by the extensibility in the $x$-direction, $1/Y_x$, and $y$-direction, $1/Y_y$ (see Methods; Supplementary Fig. 13; and Supplementary Tables 19, 20). This shows that the stiffest region (stockinette fabric in dark blue) is designed to support the radiocarpal joints and to help keep the carpal

and metacarpal bones aligned. Isotropic material (seed fabric in pink) still allows the carpometacarpal joint connecting the thumb to the wrist to move freely. Rib (green) and garter (orange) fabrics enable the fingers to extend and contract for natural motion (Fig. 4e). Importantly, knitted textiles can easily be crafted to fit any anatomy.

## Discussion

We present a picture of knitted fabric mechanics that is based on a micromechanical model of yarn. Drawing from composite theory, we have developed a mesoscale model for the relationship between bulk elastic response and local topology, entanglement, and symmetry. Our experiments and simulations demonstrate that changing the topology of stitches in a knitted fabric leads to remarkably different elastic responses, as seen in four standard types of knitted fabric. The stitch micromechanics forms the basis of a nonlinear constitutive relation that models the behavior of textiles as 2D continuous materials. The non-affine deformation of fabrics measured using digital image correlation (Fig. 4a) matches qualitatively and quantitatively with finite element simulations using our constitutive model (Fig. 4b) (see Methods and Supplementary Note 9). Our long-term goal is to automate textile metamaterial production via a pipeline that takes desired mechanical performance and, using a computational model, generates a textile with compatible local properties. This work can advance creation of non-proprietary software for designing fabric, as well as using mechanics to inform design, enabling textile engineers to tailor bespoke materials for a wide range of applications from performance sportswear[29,64] to biomedical devices[13]. With new developments in cost-effective methods to automate[2,65] and program[66] industrial knitting machines, we can build towards an open-source computational design platform that combines aspects of esthetic, functional, and mechanical design.

## Methods
### Materials and fabrication

We performed experiments on eight types of yarn that are classified in three categories: (1) two are large-gauge yarns (9-12 wraps per inch, WPI), (2) five are fine-gauge yarn (30-40 WPI), and (3) the yarn used for the therapeutic glove prototype (14-18 WPI) (see the Knitted glove prototype section).

We used Brava worsted yarn (28455-White) from KnitPicks™, which is 100% acrylic yarn, hereafter referred to as the "acrylic yarn" and 082L Pearl cotton 3/2 (color 1800-13 sapphire) from Halcyon Yarn™, which is 100% cotton yarn, hereafter referred to as the "cotton yarn." For each of the types of fabrics, we recorded the average yarn diameter within the fabric stitches as well as the average yarn lengths per stitch. Supplementary Tables 2 and 3 display the measurements for the acrylic and cotton yarn, respectively. We measured the bending rigidity, an approximate interaction potential, and the stress versus strain relationship for both types of yarn. We perform four uniaxial experiment runs on the samples to obtain the stress versus strain relationship.

We used a Taitexma™ Industrial Knitting Machine to create four types of fabrics with both the acrylic and cotton yarn: stockinette, garter, 1 × 1 rib, and seed. Each fabric sample consisted of 31 rows and columns and were made at equal tensions and stitch size settings on the machine (Supplementary Note 11). For an accurate model development, we obtained finer details of the fabric stitches. We created smaller copies of the experimental samples. We used a caliper to measure the average diameter of the yarn in situ. We then dissected them to obtain average yarn lengths per stitch for the four types of fabric.

We additionally fabricated five sets of samples (where each set contained the four types of fabrics) made from different lace weight yarns. Of the five, three yarns were from ColourMart™ : heavy lace weight alpaca mohair silk mokka 811 ecru, heavy lace weight kid mohair and silk special celeste, and 2/28NM lace weight cashmere 8l brume (beige) each referred to as "lace-weight alpaca mohair", "lace-weight blue mohair", and "lace-weight cashmere" respectively. The other two lace weights were Bambu 12 Gauge 100% Bamboo in the color 010 Rice from Silk City Fibers™, hereafter referred to as "lace-weight bamboo", and Tamm Petit 2/30 T4201 White 100% acrylic yarn from The Knit Knack Shop™, hereafter referred to as "lace-weight acrylic." These samples were fabricated on a STOLL CMS 530 HP Industrial Knitting Machine and each contained 32 rows and 32 columns. Stockinette and garter were made with a stitch size setting of 12 while rib and seed were made at size 11 (Supplementary Note 11 and Supplementary Fig. 15). All other machine parameters were kept the same. Each sample was fabricated twice with buffer regions either along its vertical or horizontal axis to aid with the uniaxial stretching experiments.

Similar to the acrylic and cotton yarns mentioned above, we measured the bending rigidity for each of these yarns and extracted the stress versus strain relationship via uniaxial experiments. Five experiment runs were performed on each sample.

To obtain the length of yarn per stitch for the lace weight samples, each sample was weighed and, using the mass density for the different types of yarn, the average length of yarn per stitch was estimated.

### Uniaxial stretching experiments

To perform the uniaxial stretching experiments, we designed a setup such that fabric samples had external forces uniformly applied to the boundary. All uniaxial stretching experiments in this work only consider loading; we do not consider the cases of unloading. We 3D printed clamps to use on both ends of the fabric samples and then had a dynamometer hooked on to one of the clamps that could be moved with a threaded rod. All components of the experiment were designed to move on guiding rails to keep everything level and prevent lateral and torsional motion. We designed the clamps with several teeth to effectively hold down both ends of the fabric sample and prevent slipping.

For each sample, we clamped the fabric on opposite ends. During the experiments, we positioned and leveled a camera above the sample. Colored pins were placed in the fabric and red points were painted on the clamps to aid with tracking during the analysis. The dynamometer was zeroed before the experiment and then incrementally moved by turning the threaded rod, applying the external force $F_x$ (or $F_y$) to the sample boundary until reaching its maximum force (30 N). Experiments were performed slowly, to approximate a quasistatic regime, stretching from a relaxed configuration to maximum extension over 1–3 min. An initialization is done for each sample where they are run through the entire experiment. This run is not included in the presented data as it is meant to break apart initial fiber connections and handling bias. Between subsequent experiment runs, the fabrics were reset to their initial resting length and briefly stretched in their transverse direction. We then waited five minutes before the next experiment run. We performed experiments along both axes of the fabrics (along its $x$- and $y$-direction).

We looked at the uniaxial response by tracking the length and waist dimensions as the external force is exerted on the boundary. For the overall bulk response, we used Fiji (https://imagej.net/Fiji) image processing software with the TrackMate plugin to track the pins and clamps on each of the sample videos and analyzed the position change of the coordinates (see Supplementary Fig. 1). The dynamometer reading was recorded using optical character recognition (OCR) on its seven segment display, ensuring stress and strain data were synchronized. Raw experimental data for the experimental runs on the acrylic, cotton, and therapeutic glove fabric samples can be found in Supplementary Fig. 3.

For the uniaxial experiments on the lace weight samples and the therapeutic glove swatches, we use an Instron Universal Testing

# Article

Machine (UTM) Model 68SC-1. We 3D printed unique clamps with teeth to fit into the machine grips and ensure no slip boundary conditions during testing. A camera is focused on the sample while stretching and two pins are placed along the transverse direction. The clamp separation is measured and the displacement is tracked by the Instron software. The samples are stretched at a rate of 0.5 mm/sec and the lace weights are stretched to 25 N while the glove samples are stretched to 30 N. The remaining experimental procedure and analysis is the same as detailed previously except that the force data was synced by matching time steps from the tracked transverse data (acquired with Fiji) and the force data (acquired with the UTM).

For one uniaxial experiment, we captured the nonaffine displacement fields throughout the entire sample under stress. We clamped the acrylic garter sample and dusted graphite powder to create a speckle pattern for tracking aid. The camera was again leveled above the sample but positioned closer to capture more detailed deformation. To analyze and track the displacement fields we use the 2D digital image correlation (DIC) MATLAB software, Ncorr (https://www.ncorr.com/).

### Yarn bending modulus measurement

Yarn has a hierarchical filamentous structure with internal stresses and friction arising from the manufacturing process that complicates determining a bending modulus through cantilever experiments. Since probe-based measurements, such as the three-point flexural test, inevitably lead to compression of the yarn's cross-section, we find that cantilever experiments yield the most consistent results, using simple approximations to the yarn shape. A schematic of the setup is shown in Supplementary Fig. 4.

Looking at four increments of yarn length ranging from 10 cm to 25 cm, we cut out five samples at each length and perform bending experiments on the yarn. For the lace weight yarns, the lengths cut were 6 cm, 9 cm, 10.5 cm, 12 cm, and 15 cm. Each sample is cut with an additional 10 cm of yarn that is adhered on a flat surface with double-sided tape. The yarn is hung off the edge of the platform and bends under its own weight due to gravity, adopting an approximately parabolic shape. A camera is positioned level to the setup and images the yarns' behavior. We apply a blur and binarize filter to the images to isolate the yarn. Taking the points that compose the yarn shape, we fit a 4th degree polynomial to find the approximate centerline of the yarn (see Supplementary Note 2).

### Yarn compressibility measurement

We used a Zwick/Roell Z010 Universal Testing Machine (UTM) to perform compression experiments on the yarn. Three yarn samples of length 20 mm (for the acrylic yarn) and 30 mm (for the cotton yarn) were compressed between a probe tip of 5 mm in diameter and a custom acrylic stage also 5 mm in diameter. The UTM probe tip was slowly lowered onto the yarn, resulting in quasistatic measurements of the restoring force as a function of probe height. A schematic of the setup is shown in Supplementary Fig. 5.

### Elastica-model simulations

To simulate the equilibrium configurations of knitted stitches, we modeled yarn as inextensible elastica with bending modulus $B$ and fixed total length $L$ per stitch. Interactions between overlapping yarn were treated with a hard-core, soft-shell model with a functional form derived from experimental measurements. Equilibrium configurations were determined by numerically minimizing the total yarn energy, given by the sum of the bending energy $E_{bend} = (B/2) \int_0^L ds \, |\partial_s \hat{\mathbf{t}}|^2$ and the core-shell interaction energy $V_{int}$, with a fixed total length constraint. To perform this numerical minimization, we represented yarn configurations as degree-5 Bézier spline curves with degrees of freedom encoded by a collection of Bézier curve control points. The resulting space curves are twice continuously differentiable with

respect to its arclength parameter $s$, and thus have continuous curvature. For more details, see Supplementary Note 4.

### Knitted glove prototype

To craft the knitted glove prototype, we used Rowan™ Baby Cashsoft Merino which is composed of 57% wool, 33% acrylic, and 10% cashmere. We used four different colors to knit the four types of fabric in the glove: blue for stockinette, orange for garter, green for rib, and pink for seed.

The fabric was knit by hand (see Supplementary Note 11) on US size 2 needles (2.75 mm in diameter) except for the stockinette regions which were knit on US size 0 needles (2 mm in diameter).

Miniature test swatches were made of each type of fabric to assist with glove design and to perform uniaxial stretching experiments on. A stockinette sample was knitted on US size 2 needles for comparison. Each sample underwent five experimental runs on the UTM. Each sample has 25 columns and 34 rows.

## Data availability

The uniaxial stretching experiment data, DIC data, bending modulus data, as well as the compressibility data used in this study can be accessed at https://hdl.handle.net/1853/73598 or on GitHub via: https://github.com/sabetta/stitch-by-stitch. Source data are provided with this paper.

## Code availability

Both the stitch-level simulation code and the FEA code are available at https://hdl.handle.net/1853/73598 or on GitHub via: https://github.com/sabetta/stitch-by-stitch.

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

## Acknowledgements

The authors thank Ali Dahaj, Daria Atkinson, James McCord, Michael Czajkowski, Paul Loveman, Peter Yunker, Robin Selinger, and Timothy Atherton for useful conversations. K.S. and A.P.C. were supported in part by the Research Corporation for the Advancement of Science Cottrell Scholar Award Grant No. CS-CSA-2020-162. M.S.D., S.G., and E.A.M. were supported by National Science Foundation Grant No. DMR-1847172. This work was supported in part by the National Science Foundation Grant No. DMS-1439786 and the Alfred P. Sloan Foundation award G-2019-11406 while the authors were in residence attending ICERM's Illustrating Mathematics program.

## Author contributions

E.A.M. designed the study, K.S. performed the uniaxial stretching and yarn compression experiments, K.S. and A.P.C. performed bending modulus experiments, K.S., S.Q., M.S.D., and S.E.G. analyzed the data, M.S.D. and S.E.G. performed the simulations, and K.S., M.S.D., S.E.G., and E.A.M. wrote the manuscript. E.A.M. designed and fabricated the therapeutic glove prototype.

## Competing interests

The authors declare no competing interests.
