## [Peer Review File · Nature Communications]

Programming mechanics in knitted materials, stitch by stitchEditorial Note: This manuscript has been previously reviewed at another journal that is not operating a transparent peer review scheme. This document only contains reviewer comments and rebuttal letters for versions considered at *Nature Communications*. Mentions of the other journal have been redacted. Additionally, parts of this file have been redacted as indicated to maintain the confidentiality of unpublished data.

REVIEWER COMMENTS

Reviewer #1 (Remarks to the Author):

This paper investigates the mechanical performance of different knitted structures based on stitch-by-stitch topological structures. The author has conducted extensive work in the field of knitting mechanics simulation. A systematic study has been conducted on this subject, with a particular focus on the experimental analysis of the characteristics of four weft-knitted structures. Preliminary findings regarding the mechanical properties of these four structures have been obtained and applied. Achieving knitting structural mechanical properties independent of yarn properties is a highly meaningful endeavor, expanding new perspectives for a profound understanding of knitted fabric performance. Here, I will present some personal insights in the hope of contributing to the paper.

Suggestion 1 :

Spandex is a widely used elastic fiber with extensive applications, ranging from the textile industry to sectors such as biomedical and automotive engineering

The author mentions "use stitch type as a way of modulating the bulk elasticity of the fabric irregardless of yarn type" to achieve the desired elastic response of a textile. I believe there are limitations to this approach. Relying solely on the elastic response provided by the fabric's knitted structure has its constraints, meaning that the fabric's structural elasticity cannot fully replace spandex. Therefore, the scope of application of this research achievement need to be further restricted and clarified.

Moreover, this paper only considers two types of yarn (acrylic and cotton), but the influence of elastic fibers on mechanical performance cannot be ruled out. If elastic fibers are incorporated into the knitted structure, and the load initially acts on the elastic fibers, would the original structural characteristics change? If changes occur, it is necessary to specify the scope of applicability of the model.

Suggestion 2 :

In qualitative analysis, it is generally believed that structures with the same configuration and the same yarn fineness but different stitch lengths show different mechanical performance, or in other words, knitted fabrics with different porosity factor have differences in their mechanical properties. This poses a challenge to the study, as it aims to determine the structural mechanical properties unrelated to yarn characteristics. The use of only two types of yarn for testing raises the question that whether these two types of yarn are representative of the majority of knitting yarns. In experimental research, two different yarns with varying fineness were employed to knit the same structure with different stitch lengths. The resulting fabrics show similar mechanical properties. This necessitates an explanation of the inevitability of the mechanical performance of materials with the same knitting structure but different yarns. Alternatively, it calls for an explanation of the mechanism and experimental basis behind the maintenance of structural performance when yarn fineness and stitch length are altered. For example, the structural performance of fabrics (under low-load conditions) is unaffected by yarn properties. It is necessary to clarify the phenomenon of significant differences in the mechanical properties of fabrics with different yarns but the same structure (as shown in Supplementary Table 10 and Supplementary Table 11).

Suggestion 3 :

The geometric models are the foundation of finite element analysis, and inaccuracies in these models can significantly impact the analysis results. In the paper, the author provides partial geometric models of stitch topologies, such as Figure 1, Figure 3, Supplementary Fig. 6, Supplementary Fig. 9, etc. What is the basis for constructing these geometric models of stitches,

and are they referenced from real stitches with different structures? While the author emphasizes that the topological structure of knitted fabrics is the primary focus of this study, it is inevitable that the geometric structure of stitches will have an impact on the mechanical properties of the fabric. It is hoped that further clarification on this matter can be included in the paper.

Reviewer #2 (Remarks to the Author):

The subject of this paper is to predict the tensile properties of weft-knitted fabrics with a combination of front and back stitches from the mechanical properties of the yarn. Additionally, the paper shows an application to a knitted glove in which the elongation properties are locally controlled by the stitches.

This could be a paper that shows the possibilities and difficulties of knitting, and the possibility of controlling them. Technically, the following points are not clear. I hope it becomes clear.

(1) Various approaches are taken at each stage of mechanical simulation of weft knitted fabric. The stages are the Initial Loop model (unloaded state), the physical model of yarn in weft knitted fabrics (including measurement of properties), the calculation method of response to external force or displacement, and its verification by experiment [1-7]. If you have not yet cited a paper, you should cite it and state the novelty and usefulness of this paper.

[1] Sha, S., Geng, A., Gao, Y., Li, B., Jiang, X., Tao, H., ... & Chen, Z. (2021). Review on the 3-D simulation for weft knitted fabric. *Journal of Engineered Fibers and Fabrics*, 16, 15589250211012527.

[2] Y. K. KYOSEV, FEM and its application to textile technology, in *Simulation in textile technology*, 172-221, 2012, Woodhead Publishing

[3] R. B. Ramgulam, Modeling of knitting, in *Advances in knitting technology*, 48-85, 2011, Woodhead Publishing

[4] Htoo, N. N., Soga, A., Wakako, L., Ohta, K., & Kinari, T. (2017). 3-Dimension Simulation for Loop Structure of Weft Knitted Fabric Considering Mechanical Properties of Yarn. *Journal of Fiber Science and Technology*, 73(5), 105-113.

[5] Kaldor, J., James, D.L., Marschner, S.: Simulating knitted cloth at the yarn level. In: *Proceedings of SIGGRAPH 2008*. Held in Los Angeles, California, Aug 2008 (2008)

[6] Wadekar P, Perumal V, Dion G, et al. An optimized yarn level geometric model for Finite Element Analysis of weft knitted fabrics. *Comput Aided Geom Des* 2020; 80: 101883.

[7] Ru, X., Wang, J. C., Peng, L., Shi, W., & Hu, X. (2023). Modeling and deformation simulation of weft knitted fabric at yarn level. *Textile Research Journal*, 93(11-12), 2437-2448.

(2) There are countless variations of knits and threads, so it is necessary to limit them. This paper focuses on basic weft knitting that combines front and back stitches. Among these types of knitted fabrics, elongation simulations of plain knit (Stockinette) and rib knit have already been performed.

The initial state modeling idea in this paper is the same as above Refs and I think it is not novel.

(3) This paper incorporates tension and bending of yarn. In predicting fabric mechanics from yarn, the following are known as the necessary physical properties of yarn. Micro-level fabric deformation modes: (a) inter-yarn slip, (b) inter-yarn shear, (c) yarn bending, (d) yarn buckling, (e) intra-yarn slip (inter-fibre friction), (f) yarn stretching, (g) yarn compression, (h) yarn twist [8].

[8] Duhovic, M., & Bhattacharyya, D. (2006). Simulating the deformation mechanisms of knitted fabric composites. *Composites Part A: Applied Science and Manufacturing*, 37(11), 1897-1915.

Of course, a model that incorporates everything is not realistically possible. For example, the Hearle et. al [9] provides an idea of what kind of model corresponds to what kind of thread.

[9] Hearle, J. W. S., Potluri, P., & Thammandra, V. S. (2001). Modelling fabric mechanics. *Journal of the Textile Institute*, 92(3), 53-69.

Please clearly explain whether the assumptions in this paper match the physical properties of the thread in the verification experiment. In particular, it is questionable whether it is appropriate to

ignore friction and tension.

(4) In the xx (course) direction in Figure 2, except for plain knitting, the experiments and calculations are different, but I am unable to understand the cause of this difference. For example, if you experiment with a monofilament knitted fabric with high compression stiffness, and/or with a structure where the contact points are fixed, I think the reason will become clearer.

(5) Additional comments

About yarn compression stiffness measurement;

Compression is performed using a plane compressor. I believe that the contact during knitting due to yarn contact will change from point contact to area contact. It is unclear whether the same modulus can be used in the model in this paper.

About yarn bending stiffness measurement;

The bending stiffness of yarn was measured by a cantilever method. I think the curvature of the cantilever is too small compared to the curvature of the yarn in knitted fabrics. Please state why the measurements are valid.

Reviewer #3 (Remarks to the Author):

The presented manuscript is interesting from the point of view of theoretical modeling, but the practical application is unconvincing. The authors present many examples of functional textiles in the introduction, but the work presented by them has little connection with the cited articles. The introduction must be strengthened by providing a detailed analysis of the knitted structure models that have been developed. It is also necessary to discuss scientific works examining the mechanical behavior of knitted fabrics. There are many developed theoretical models of the knitted loop that allow accurate estimation of the length and shape of the loop, which allows to predict the properties and mechanical behavior of the knitted fabric. List of references and the introductory text must be very close related to the investigated topic.

When describing the object of research, the authors must use standardized names of knitted patterns (in this case, they provided only one correct name - rib) and correct expressions. For example, instead of "knitted fabric" (when referring to the specific arrangement of the loops in the knitted fabric) it must be used "knitting pattern" or "pattern"; "stitch gauge" must be "stitch density", "yarn per stitch" - "loop length", etc.

I would not agree that when modeling the mechanical behavior of knitwear it is correct to exclude the influence of friction at the points of intersection of yarns, and to analyze only the bending and compression forces. Because the friction between the yarns at the points of intersection of the stitches is one of the factors that help to maintain the specific shape of the stitch. Therefore, friction must be considered in the simulations.

The authors mentioned that they "we use the elastica model", however, elastica model can give reasonable accuracy only when using woolen yarns.

For the production of the prototype, the authors chose hand knitting. The authors claim that "The knitting machine is ideal for ensuring uniform tension throughout the sample; however, it comes at the expense of not guaranteeing uniform stitch size between types of fabrics. Hand knitting, in contrast, cannot guarantee uniform tension throughout the sample, but provides greater control of stitch size even while altering the pattern of the knit and purl stitches". But this is an incorrect statement, since subjective factors are at work during manual knitting, which in no way guarantee the same length of the knitted loops. Even more, the principle of the glove prototype knitting is very unusual, it is not clear why the authors did not choose a traditional seamless knitting method. Authors present that "We targeted 20 - 30 mm Hg (compression) for the hand of the specific wearer (...). We have achieved this comparable pressure without the use of elastane (!)." It is interesting, how did they achieve this level of compression (this part of the glove must be and how did they measure it? This part of the glove must be correspondingly smaller in circumference than the wrist of the hand; without elastomeric yarn, such a product is generally difficult to apply to the limb.

In order to improve the value of the submitted work, I recommend consulting with a textile

(knitting) engineering specialist, and also familiarizing yourself with published works that examine real models of knitted loop and properties of knitted structures.

Reviewer #4 (Remarks to the Author):

This paper explores the relationship between knitted fabric stitch topology and resulting knot elasticity through experiments and simulation. The authors aim at constructing a constitutive model for the nonlinear bulk response of such fabrics. The goal is to untangle the relationship between stitch pattern and mechanical response for the 4 most common knitted fabrics: stockinette, garter, rib and seed. The authors measured the elastic response of each of the four fabrics in a series of uniaxial stretching experiments and simulations, using different types of yarn with different mechanical properties. The simulations show that in the low/high stress regime, bending energy/compression energy is the dominant contributor to elastic response and in the high stress regime. By breaking the problem into a concatenation of two types of stitches it is shown that odd symmetry connecting yarn segments can be of order ten times softer than compared to even symmetry connecting yarn segments.

The presentation of the paper is overall very good and the results are innovative and significant towards understanding of the relation between topology of filaments and mechanics of filamentous matter in general. Therefore, I recommend this paper for publication.

Some more detailed comments are below:

In the caption of Fig.2 it is said that the experimental data are shown in the translucent regions. These are not clear in the figure. The data would be more visible if either the figures are larger or if the lines are thicker.

In the same also figure, dashed lines are mentioned in the caption, but they are not visible in the figure.

The caption refers to the patterns by their knit names, but it would be helpful if in parenthesis there was a mention of the symbol used in the figure for each of the knits. The figure itself shows the knits, but not the names and the reader has to look back and forth in the paper to make sense of the caption.

Finally, in the main text it is said that Fig.2 shows results for different types of yarn. I think it is meant as an average. A few words here about the characteristics of different type of yarn as they are shown in the Supplementary information would be useful.

Responses to Referees

Dec 2023

We thank the referees for their helpful comments. We realize that the journal is meant for a broad readership and we made efforts to clarify how our work is situated in the field and amongst past studies. We appreciate receiving comments from different perspectives and will use them to make the paper more understandable to a more general audience. We hope, after careful consideration and revision, that this manuscript remains of interest to Nature Communications. We would like to address the comments of the referees as follows (our comments are in blue), where Refs 1, 2 3, and 4 are the corresponding referees from our Nature Communications submission [REDACTED]

Changes made since ALL referees have reviewed:

1. Some reviewers had concerns about the scope of our conclusions for different yarn types, and if the yarn types we studied in this paper represent the body of yarns used in fabric manufacturing. To address this issue, we have made the following additions:
 - We have manufactured five additional set of samples in lace weight (one of each fabric type in acrylic, blue mohair, cashmere, alpaca silk mohair, and bamboo yarn) and conducted force-extension experiments on all of the new samples. We additionally conducted bending modulus experiments for each lace-weight yarn and included the results in Supplementary Table 1.
 - We characterized all of the stress-strain behavior of the new samples using the constitutive relation we proposed in the paper, described in Supplementary Tables 13 and 14.
 - We conducted a comparison of the rigidities of the fabric types in all fabric samples, both the additional new samples and original samples included in the paper. This comparison is now included as Figure 2c and Supplementary Fig. 11. From these figures, we can see clear groupings of the (normalized) rigidities of each fabric type, supporting our argument that the topology of the stitches affects the rigidity regardless of yarn type used.
 - To summarize the new experimental procedure for these samples made on a different knitting machine, we have added text to the Methods section and the Supplementary Text (Notes 1 and 2) as well as included images of the new experiment setup on the Universal Testing Machine in Supplementary Figure 1 as subplots c and d.
2. Multiple reviewers had questions about friction. To add clarity to the paper, we have added a section to the SI devoted entirely to a discussion of friction, Supplementary Note 4.7. We have addressed reviewer concerns individually below, but would like to give a short summary of our main points here:
 - We do not see frictional affects on the elasticity of the knit fabrics when we iteratively repeat extension experiments, as seen in Supplementary Fig. 3.
 - Our stitch-level simulations are not dynamic. Though friction can provide an important stabilizing effect, we are using static simulations that will not unravel without friction. We are also not simulating sections of fabric that might come into contact with each other, which is an interaction where friction would be very important.
 - Friction cannot be incorporated in an energy minimization simulation. The current simulation calculates the bending and compression energy at every step and minimizes this energy through

the in-built SciPy minimize function, as described in Supplementary Note 4.1. A dissipative term, like friction, cannot be incorporated into this framework. We are currently working on a new simulation that uses gradient descent and thus can incorporate friction, but it is unlikely this simulation will be finished within the next year.

- Prior research shows that energy lost to friction is a very small contribution, at most the third-largest contribution for knit fabrics made of incompressible yarn (Duhovic & Bhattacharyya 2006). Since we are using compressible yarn with an appreciable compression energy contribution (which is not present for incompressible yarn), we estimate that friction has a similarly small energy contribution in our experiments.
- Without friction, we already have very good agreement between experimental and simulated stress-strain data. Our stress-strain simulation results are able to replicate the shape of both the linear and non-linear elastic responses of knit fabrics, which has yet to be achieved for knits made of compressible yarns. Even the most recent work modeling knits with friction (Ding et al., arXiv:2307.12360v2, Figures 4 & 5) does not get as good agreement to experimental results over the entirety of the stress-strain curve.

3. We have made a few changes to the manuscript to reflect points of clarity brought up from all of the referees, as follows:

- We have better identified the scope of yarns our work applies to and that we are not considering elastic yarn such as elastane or nylon.
- We further described how constraining the length of yarn per stitch accounts for manufacturing tension and included a new figure, now Supplementary Figure 9.
- We clarified that the FEA applies our constitutive stress-strain relation to a 2D sheet of fabric and the FEA does not directly consider the local microstructure of the stitches within the fabric.
- We described how the stitches in the stitch-level simulations are initialized.
- We described the data that was rendered for stitch pattern renderings in Figures 1, 3, S6, S8, S9 (new), and S10 (old S9).
- We better identified how our stitch level simulations are situated within previous work and that the novelty of our research is in how these simulations were used to study the mechanical effects of stitch topology, not the simulations themselves.
- We have added additional names for the common knitting terms used in our paper, including the knit stitch (“front stitch”), purl stitch (“back stitch”), length of yarn per stitch (“loop length”), stitch gauge (“stitch density”), and garter (“links-links”).
- We have cited Supplementary Table 2 (previously Supplementary Table 1) to support our claim that hand-knitting leads to more consistent length of yarn per stitch over multiple fabric types in comparison to machine knitting.
- We have described how we estimated the compression that the glove prototype applies to the wrist.
- We have included the fabric type names to the legends for Figures 2, S2, and S12 (old S10).
- We defined elastica within the Supplementary Materials.

4. Other changes we’ve completed:

- We have added additional citations to both the main paper (four citations) and the Supplementary Information (11 citations) since our Nature Communications submission. Since our original Submission, we have added 15 citations to the main text and 26 citations to the Supplementary Information. At this intersection of metamaterials, textile engineering, and computer graphics there is a wealth of research in all of these fields and, unfortunately, we are limited in the number of citations we can make in the main text.
- We have removed the reference to dashed lines in the caption of Figure 2.

- Supplementary Fig. 3 was updated with new raw experimental data for the glove prototype samples.
- We've updated Supplementary Fig. 12 (previously 10) with the new experimental data on the glove prototype samples.
- We've included a new figure, Supplementary Fig. 14, that compares fabrics made of different stitch size settings on the industrial knitting machine.
- We performed bending modulus experiments on the wool-blend yarn used to fabricate the glove prototype samples and reported the value in Supplementary Table 1.
- We included experiments and fits on an additional stockinette sample made from the wool-blend yarn with 2.75 mm (US size 2) knitting needles (Fig. 2c, Supplementary Figs. 3 and 11, and Supplementary Tables 6, 19, and 20).
- We spelled out numbers listed in the text numerically (not including references, figure numbers, measurements, etc.).
- We measured the area per stitch for all fabric samples and included this data in Supplementary Tables 5 and 6.

With the addition of new Supplementary Figures and Tables, we have redone all references to the SI materials in the main text. The conversion goes as follows:

Original ██████ SI Figure Number	Original NatComm SI Figure Number	New SI Figure Number
5	1	1
6	2	2
-	3	3
1	4	4
2	5	5
7	6	6
-	7	7
-	8	8
-	-	9
8	9	10
-	-	11
3	10	12
4	11	13
-	-	14

Original [REDACTED] SI Table Number	Original NatComm SI Table Number	New SI Table Number
-	-	1
1	1	2
2	2	3
7	3	4
-	-	5
-	-	6
8	4	7
-	5	8
3	6	9
-	-	10
5	7	11
-	-	12
-	-	13
-	-	14
-	8	15
-	9	16
4	10	17
6	11	18
9	12	19
10	13	20

Additional global changes [REDACTED]

1. We have made a few changes to the manuscript to reflect points of clarity brought up from all of the referees, as follows:

- Within the main text, we have included text to clarify our use of simulation, including differences between how simulations are approached in the physics/engineering community versus approaches of the graphics community.
- Within the Discussion section of the main text we added the line “*Drawing from composite theory...*” on line 205.
- In the numerical model section we have made clear which yarn contributions we are considering.
- We have more clearly situated our work relative to past studies.
- We have added several sections and figures in the SI breaking down separate contributions to the total energy due to yarn bending and compression, as well as an observed jamming behavior, and instability observed in seed.
- We have reorganized the Supplementary Information to better reflect the organization of the main text. [REDACTED]

2. Other changes we’ve completed:

- A slight update to the experimentally derived bending modulus for the acrylic yarn.
- A simulation error involving the calculation of bending energy was corrected.
- The comparison between theory (“RS model”) and experimental measurements of Young’s moduli has been re-plotted to better show the good agreement over two decades.
- Reported error for experimental measurements of Young’s moduli for the original cotton and acrylic samples.

Referee #1

This paper investigates the mechanical performance of different knitted structures based on stitch-by-stitch topological structures. The author has conducted extensive work in the field of knitting mechanics simulation. A systematic study has been conducted on this subject, with a particular focus on the experimental analysis of the characteristics of four weft-knitted structures. Preliminary findings regarding the mechanical properties of these four structures have been obtained and applied. Achieving knitting structural mechanical properties independent of yarn properties is a highly meaningful endeavor, expanding new perspectives for a profound understanding of knitted fabric performance. Here, I will present some personal insights in the hope of contributing to the paper.

Suggestion 1:

Spandex is a widely used elastic fiber with extensive applications, ranging from the textile industry to sectors such as biomedical and automotive engineering. The author mentions “use stitch type as a way of modulating the bulk elasticity of the fabric irregardless of yarn type” to achieve the desired elastic response of a textile. I believe there are limitations to this approach. Relying solely on the elastic response provided by the fabric’s knitted structure has its constraints, meaning that the fabric’s structural elasticity cannot fully replace spandex. Therefore, the scope of application of this research achievement need to be further restricted and clarified. Moreover, this paper only considers two types of yarn (acrylic and cotton), but the influence of elastic fibers on mechanical performance cannot be ruled out. If elastic fibers are incorporated into the knitted structure, and the load initially acts on the elastic fibers, would the original structural characteristics change? If changes occur, it is necessary to specify the scope of applicability of the model.

We agree that the addition of elastane has a large impact on the mechanics of knit fabrics, and investigating that effect is important. To understand how elastane changes the behavior of knit fabrics, we must first understand how knit fabrics behave without elastane. This paper is an attempt to do just that. We believe that incorporating the effect of elastane is beyond the scope of this first paper, but that the results we report here will be useful in quantifying the effects of incorporating elastane in future research. It is also important to note that not all knit fabrics use elastane, and the results in this paper apply to those fabrics.

We are inspired by the manipulation of knit and purl stitches in hand-knitting projects, where the techniques and materials employed in industrial manufacturing are not as often used. Hand-knitters are more limited in the materials they have access to and use stitch patterning as their primary tool for modulating fabric elasticity within a single garment. The purpose of this paper is to push this “at-home textile creation” technique; here, we harness simulations and modeling to understand the mechanisms by which this behavior is achieved. We fully acknowledge that one cannot achieve fully comparable results by only altering stitch patterns, but the drastic changes in elastic response achievable by just small changes in topology can expand the realm and applicability of knitted textiles across scales (as mentioned in our introduction).

For clarity, we have added the bolded section to the main text:

*“Here, **inspired by the design of hand-knit garments**, we study how the mechanical behavior of weft knitted fabrics is encoded by the topology of their stitches as a first step towards creating a design tool for programmable textile metamaterials.”*

In preliminary research, we see two linear regimes for fabrics made of extensional yarn: a pattern-dominated low-force regime and an intermediate-force, yarn-stretching regime. We have some (unpublished) data that shows these two regimes under a different context than what is being presented in this paper. That data was collected on a very different experimental setup and is not comparable quantitatively to the data presented here. We agree that a full characterization of the knitting design space requires incorporation of yarn extensibility, but given that we are focused on the lower-force linear regime, which is determined by stitch topology and bending modulus, we leave the addition of extensibility for future work. Our work advances use of topology and geometry for designing aspects of fabric response; engineering response through constituent material properties represents another dimension of the “design space” that it is worth investigating in the future, but would be built upon the foundation presented here.

To address this comment and further specify the scope of our results, we have added the following text to the main paper:

“To maximise extensibility, manufacturers reduce the amount of natural fibers used in the fabric and increase the amount of elastane and/or other elastomeric fibers. Our goal is to use stitch type as a way of modulating the bulk elasticity of fabrics made of inelastic yarn, irregardless of fiber composition, so that the desired elastic response of a textile can be achieved with natural and/or biodegradable fibers and without synthetic materials. Recent research has shown that a broad range of synthetic materials can degrade when in contact with skin secretions, which increases the potential for dermal absorption of compounds within those fibers (Abafe et al 2023).”

Suggestion 2:

In qualitative analysis, it is generally believed that structures with the same configuration and the same yarn fineness but different stitch lengths show different mechanical performance, or in other words, knitted fabrics with different porosity factor have differences in their mechanical properties. This poses a challenge to the study, as it aims to determine the structural mechanical properties unrelated to yarn characteristics. The use of only two types of yarn for testing raises the question that whether these two types of yarn are representative of the majority of knitting yarns.

To address this concern, we’ve included stress-versus-strain experiments on five additional sets of samples made from different yarn types: a mohair silk blend, cashmere, an alpaca mohair silk blend, bamboo, and a lace-weight acrylic. This further expands our results to include more yarn blends and incorporates more natural fibers. All of these additional samples are made of lace-weight yarn, a more commonly used industry weight for clothing. Additionally, we now include a comparison of length of yarn per stitch and area per stitch in Supplementary Tables 2, 3, 5, and 6, which together give a measure of porosity. To account for porosity affects, we display the normalized rigidities of all fabric samples (the two original acrylic and cotton samples, the glove prototype samples, and the five new lace-weight samples) in the new Fig. 2c, where differences in porosity and yarn bending modulus are divided out. In this plot, we can see groupings by fabric types, as displayed by the ellipses. Each ellipse is centered at the average normalized rigidity for it’s associated fabric type, and the width of the ellipse represents the standard deviation.

We have also provided a comparison of hand-knit stockinette using US size-0 and US size-2 needles. You can see a dramatic difference in the Young’s moduli of these two fabrics (factor of five in the x -direction and factor of two in the y -direction) in Supplementary Table 20.

In experimental research, two different yarns with varying fineness were employed to knit the same structure with different stitch lengths. The resulting fabrics show similar mechanical properties. This necessitates an explanation of the inevitability of the mechanical performance of materials with the same knitting structure but different yarns. Alternatively, it calls for an explanation of the mechanism and experimental basis behind the maintenance of structural performance when yarn fineness and stitch length are altered. For example, the structural performance of fabrics (under low-load conditions) is unaffected by yarn properties. It is necessary to clarify the phenomenon of significant differences in the mechanical properties of fabrics with different yarns but the same structure (as shown in Supplementary Table 10 (now 17) and Supplementary Table 11 (now 18)).

The acrylic and cotton samples do show similar elastic behavior as a function of fabric type. This supports our conclusion that the topology of the stitches affects overall fabric elasticity. We theoretically derive this result in Supplementary Note 8 and show that the connection between a knit and purl stitch is approximately 10 times softer than that between two knit stitches, regardless of yarn type. We have also performed additional experiments on other yarn types, showing that each fabric type has a characteristic, pattern-dependent response when comparing the *relative* stiffnesses of each fabric made from the same yarn. This additional data is displayed in both Fig. 2c and Supplementary Figure 11.

Fabric elasticity is dependent on many fabric and yarn properties, and some of these factors affect each direction of the fabric differently. For example, cotton garter in the x -direction is nearly five times as elastic as acrylic garter in the linear regime, yet the rigidity in the y -direction only differs by a factor of two (Supplementary Tables 17 and 18). The additional new lace-weight data emphasizes the need for additional research in this area. Though the manufacturing conditions are the same over all yarn types, there is a wide range of mechanical behavior even over samples made at the same yarn weight. For example, the stockinette sample for lace-weight acrylic yarn is nearly an order of magnitude more rigid than the stockinette sample made of lace-weight bamboo yarn (Supplementary Table 14). It is also interesting to consider the standard

deviations of the rigidities of each fabric type, shown in the new Fig 2c and Supplementary Fig. 11. As displayed in the size of the ellipses, stockinette has a much larger standard deviation in the rigidity than the other fabric types. We are currently working on a second paper that investigates the affect of different yarn types and stitch sizes on fabric elasticity, but we believe this investigation is beyond the scope of this paper, which focuses on how topology alone affects fabric elasticity.

Suggestion 3:

The geometric models are the foundation of finite element analysis, and inaccuracies in these models can significantly impact the analysis results. In the paper, the author provides partial geometric models of stitch topologies, such as Figure 1, Figure 3, Supplementary Fig. 6, Supplementary Fig. 9 (now Supplementary Fig. 10), etc. What is the basis for constructing these geometric models of stitches, and are they referenced from real stitches with different structures? While the author emphasizes that the topological structure of knitted fabrics is the primary focus of this study, it is inevitable that the geometric structure of stitches will have an impact on the mechanical properties of the fabric. It is hoped that further clarification on this matter can be included in the paper.

The reviewer is absolutely correct that the geometric structure of stitches has an impact on mechanical properties of the fabric. The geometric structure is incorporated within our RS model and yarn-level simulations, but is not directly incorporated into the FEA. Geometric modeling comes into play with our RS model. The RS model takes in geometric parameters determined by the yarn-level simulations. This then informs the the constitutive relationship used in FEA.

The FEA takes in the constitutive relationship between stress and strain. The form of the constitutive stress-strain relation is based on the underlying micro-mechanical structure, but the relevant elastic constants are taken from experimental stress-strain data. The FEA itself is not modelling the micro-structure of the stitches, only applying the constitutive relation to a large-scale sheet of the same dimensions as our experimental samples. Thus, the geometric shape of the stitches given by the stitch-level simulations are not inputs or *directly* considered by the FEA. Further FEA details are included in Supplementary Note 9.

We have added the following to the main text:

“In this work, our goal is to study knit fabrics from three different types of models: a minimal model of yarn-level simulation at the microscopic level, a constitutive model at the textile level, and our “Reduced-Symmetry” model at the intermediate level to unite these two points of view.”

For clarity, we have added the bolded sections to the following text of the main paper:

*“The bulk constitutive model can nonetheless well-approximate the full deformation of a finite swatch of knitted fabric, as illustrated in Fig. 4a,b, where we compare the x-component of the displacement field of a sample of garter fabric stretched in the y-direction (measured using digital image correlation, DIC) against a finite element analysis (FEA) that applies our constitutive model to a **two-dimensional sheet** with more realistic boundary conditions **without directly considering the local microstructure** (Supplementary Note 9).”*

We have also added to Supplementary Note 9:

“The points \mathbf{r} and \mathbf{R} lie on two-dimensional triangular meshes with the same topology (no re-meshing is performed during the calculation),” where points \mathbf{r} are the points inside the FEA modelled fabric before deformation and \mathbf{R} are the points after deformation.

The figures the referee cites show geometric models that are outputs of the stitch-level simulations. The initial input geometry of these stitch-level simulations are based on real knit and purl stitches. The simulations take a general form of the knit or purl stitch and apply the specific yarn and fabric properties. The simulation then changes the precise shape of the yarn within the stitch in order to minimize the total energy (find the mechanical equilibrium configuration). The general geometry of the stitch remains the same. For example, a knit stitch cannot change into a purl stitch. Details on the stitch-level simulation method are described in Supplementary Note 4.1.

We have added the following text on simulation initialization to Supplementary Note 4.6:

“In the simulation, manufacturing tension is relevant during initialization of the fabric before it is stretched or deformed in any way. Others have simulated the actual knitting process (Duhovic & Bhat-tacharyya 2006), implemented a shrinking factor that reduces the arc-length of segments of yarn until the fabric settles into a rest-state (Kaldor et al. 2008), or taken a picture of a physical sample and used that geometry as an input of the simulation that is then relaxed to near force-balance (Sperl et al. 2022). Many don’t consider tension at all (Ru et al. 2023). Of these initialization strategies, the method we use is closest to that of Sperl et al. (Sperl et al. 2022); we start with a input geometry inspired by the actual geometry of the stitches within the fabric. We then impose constraints and yarn properties. The length constraint, which fixes the length of yarn per stitch, is how we control how tightly the stitches are manufactured. Once these input properties are imposed, we allow the simulation to find the minimum energy configuration that fulfills these constraints. In Supplementary Fig. 9, we show how we can control the tightness of the stitches by changing the length of yarn per stitch.”

For clarity, we have added to captions that include outputs of the stitch-level simulations:

- Fig. 3: *“The renderings in (a-j) are repeated unit cells of sample stitch-level simulation outputs.”*
- Supplementary Fig. 6: *“These diagrams were created using sample outputs of the stitch-level simulations.”*
- Supplementary Fig. 8: *“These renderings (c,d) were made using the outputs of the seed simulations.”*
- Supplementary Fig. 10: *“These renderings we done using sample outputs of the stitch-level simulations.”*

The exact geometric structure of the stitches within the fabric is dictated by the manufacturing and yarn properties. As mentioned in response to your previous suggestion, we are working on a second paper to characterize this dependence.

Referee #2

The subject of this paper is to predict the tensile properties of weft-knitted fabrics with a combination of front and back stitches from the mechanical properties of the yarn. Additionally, the paper shows an application to a knitted glove in which the elongation properties are locally controlled by the stitches. This could be a paper that shows the possibilities and difficulties of knitting, and the possibility of controlling them. Technically, the following points are not clear. I hope it becomes clear.

(1) Various approaches are taken at each stage of mechanical simulation of weft knitted fabric. The stages are the Initial Loop model (unloaded state), the physical model of yarn in weft knitted fabrics (including measurement of properties), the calculation method of response to external force or displacement, and its verification by experiment [1-7]. If you have not yet cited a paper, you should cite it and state the novelty and usefulness of this paper.

[1]Sha, S., Geng, A., Gao, Y., Li, B., Jiang, X., Tao, H., ... & Chen, Z. (2021). Review on the 3-D simulation for weft knitted fabric. *Journal of Engineered Fibers and Fabrics*, 16, 15589250211012527.

This review paper does a good job of summarizing numerical techniques to model knitted fabrics. We have added a citation to this paper to the Supplementary Materials. Through this review, we found the following paper: Abghary MJ, Hasani H, Nedoushan RJ. Numerical simulating the tensile behavior of 1×1 rib knitted fabrics using a novel geometrical model, *Fibers Polym* 2016; 17(5): 795–800. This paper compares simulation results to experimental measurements and thus has great relevancy to our work and has been added as a citation in the main text.

[2]Y. K KYOSEV, FEM and its application to textile technology, in *Simulation in textile technology*, 172-221, 2012, Woodhead Publishing

We have added a citation to this work in the Supplementary Materials.

[3] R. B. Ramgulam, Modeling of knitting, in Advances in knitting technology, 48-85, 2011, Woodhead Publishing

We have included a citation to this work in the Supplementary Materials.

[4] Htoo, N. N., Soga, A., Wakako, L., Ohta, K., & Kinari, T. (2017). 3-Dimension Simulation for Loop Structure of Weft Knitted Fabric Considering Mechanical Properties of Yarn. Journal of Fiber Science and Technology, 73(5), 105-113.

We have added a citation to this paper in the Introduction of the main text.

[5] Kaldor, J., James, D.L., Marschner, S.: Simulating knitted cloth at the yarn level. In: Proceedings of SIGGRAPH 2008. Held in Los Angeles, California, Aug 2008 (2008)

This paper is already cited in the main text and the Supplementary Information.

[6] Wadekar P, Perumal V, Dion G, et al. An optimized yarn level geometric model for Finite Element Analysis of weft knitted fabrics. Comput Aided Geom Des 2020; 80: 101883.

This paper is already cited in the main text and the Supplementary Information.

[7] Ru, X., Wang, J. C., Peng, L., Shi, W., & Hu, X. (2023). Modeling and deformation simulation of weft knitted fabric at yarn level. Textile Research Journal, 93(11-12), 2437-2448.

We have added a citation to this paper in the Supplementary Materials.

(2) There are countless variations of knits and threads, so it is necessary to limit them. This paper focuses on basic weft knitting that combines front and back stitches. Among these types of knitted fabrics, elongation simulations of plain knit (Stockinette) and rib knit have already been performed. The initial state modeling idea in this paper is the same as above Refs and I think it is not novel.

In the main text, we have clarified the role of the three different types of models used in this paper:

“In this work, our goal is to study knit fabrics from three different types of models: a minimal model of yarn-level simulation at the microscopic level, a constitutive model at the textile level, and our “Reduced-Symmetry” model at the intermediate level to unite these two points of view.”

We further contextualize our stitch-level simulations by adding the following to the main text:

“Stitch-level simulations (also known as loop modeling) have been of interest to a variety of fields, including textile engineering and metamaterials. Current simulations typically have at least one of three primary limiting constraints: they do not consider compressible yarn (Abghary et al. 2016, Poincloux et al. 2018, Duhovic & Bhattacharyya 2006), they only consider one fabric type (Abghary et al. 2016, Duhovic & Bhattacharyya 2006, Htoo et al. 2017, Abel et al. 2012, Poincloux et al. 2018), or they only compare simulation to experimental results for visual fidelity and not mechanical response (Ru et al. 2023, Kaldor et al. 2008). Our simulation method considers all three of these factors to investigate the role of stitch topology on the mechanical behavior of knit fabrics.”

We have additionally included the following to the main text:

“By implementing a minimal model in simulations, we can determine the key ingredients that contribute to the different mechanical behavior of different fabric types so that our results can be efficiently utilized in the fields of mechanical metamaterials and extreme mechanics.”

Our stitch-level simulations are not novel; what is novel is how we are using them. We utilize the simulations to investigate how the topology of the stitches affects the fabric elasticity. Instead of asking about the mechanics of a single loop or stitch pattern, we generate a theoretical model to go beyond a single loop and what a single simulation can tell us about mechanics. It is the combination and interpretation of simulations that is unique, not the simulation by itself.

Using our simulation framework, we are able to look at multiple stitch patterns and compare the elasticity. We explicitly consider the fabric topology in a minimal model where we include bending energy and yarn-yarn interactions. We can equally well consider models with more levels of detail within the same framework. We chose this model expressly because it enables us to look at effects of topology via yarn-yarn interactions and geometry via bending. Additions to this model would enable us to study other phenomena not considered in this paper. We reproduced the behaviors of stockinette and rib that people have seen before, and we are able to compare them to two additional, different fabric types.

(3) This paper incorporates tension and bending of yarn. In predicting fabric mechanics from yarn, the following are known as the necessary physical properties of yarn. Micro-level fabric deformation modes: (a) inter-yarn slip, (b) inter-yarn shear, (c) yarn bending, (d) yarn buckling, (e) intra-yarn slip (inter-fibre friction), (f) yarn stretching, (g) yarn compression, (h) yarn twist [8]. [8]Duhovic, M., & Bhattacharyya, D. (2006). Simulating the deformation mechanisms of knitted fabric composites. *Composites Part A: Applied Science and Manufacturing*, 37(11), 1897-1915.

Of course, a model that incorporates everything is not realistically possible. For example, the Hearle et al [9] provides an idea of what kind of model corresponds to what kind of thread. [9] Hearle, J. W. S., Potluri, P., & Thammandra, V. S. (2001). Modelling fabric mechanics. *Journal of the Textile Institute*, 92(3), 53-69. Please clearly explain whether the assumptions in this paper match the physical properties of the thread in the verification experiment. In particular, it is questionable whether it is appropriate to ignore friction and tension.

We address a majority of the energy considerations described by Hearle et al, described in their Table 1 for compact yarns and section 4.3, Knits. The changes in bending energy caused by yarn flattening is a higher order correction and we agree that it should be systematically investigated in the future. For non-spun yarn, there is preliminary work considering this issue by Pedro Reis at EPFL (Grandgeorge et al. 2021). We will specifically address questions about friction and tension in the response below.

Duhovic & Bhattacharyya analyze these energy modes in the context of incompressible yarn, which has a very different mechanical response in accordance with the yarn’s different composition. For example, Duhovic & Bhattacharyya state that yarn compression is “relatively insignificant,” which may be true for incompressible yarns but is not true for compressible yarn. With their incompressible yarn, they conclude that bending is by far the dominant energy contribution for the entirety of the fabric-extension experiment. At low strain, they conclude that torsion is the second-highest energy contribution. At high strain, they conclude that yarn stretching (axial tension) and then yarn contacts become the second-largest energy contributions. At no point during the extension experiments is friction the second-largest energy contribution.

For our compressible yarn, compression energy becomes a significant energy contribution. We address the contributions of yarn stretching and twist in the main text, which Duhovic & Bhattacharyya identified as the next-most significant energy contributions. In comparison to the sum of bending and compression energy for our yarn, both yarn stretching and twist are very small. By the conclusion of Duhovic & Bhattacharyya, friction would be even smaller. Yarn buckling is extremely improbable considering we have a conventionally spun yarn and we aren’t compressing the fabric. We are not considering the yarn at the fiber level, which eliminates methods of accounting for intra-yarn slip.

To address concerns about friction, we have added the following text as Supplementary Note 4.7:

“Friction is often included in dynamic simulations of knitted fabrics (Kaldor et al. 2008, Cirio et al. 2017, Duhovic & Bhattacharyya 2006). We do not include friction in our stitch-level simulations, primarily because our simulations are static. Friction can not be incorporated into an energy minimization scheme. Including a dissipative term such as friction into a static simulation is not supported by both the general simulation method and the specific way we minimize the energy.

For each set of stitch cell dimensions, we iteratively change the shape of the stitch to find the minimum energy configuration (Sokolnikoff 1956), as previously described. We are not continuously stretching the stitch cell. Each set of stitch cell dimensions is a single simulation, unconnected to other simulations of different stitch cell dimensions. Static simulations of this kind well suit our purposes to use simulations to investigate the role of topology on fabric mechanics. By finding the mechanical equilibrium point of the stitch cell for each set of given dimensions, we well represent the mechanics of our experiments. Our stress-strain simulation results are able to replicate the shape of both the linear and non-linear elastic responses of knit fabrics, which has yet to be achieved for knits made of compressible yarns.

As mentioned in Supplementary Note 4.1, we use the Sequential Least Squares Programming (SLSQP) method in the `scipy.optimize` Python package (<https://docs.scipy.org/doc/scipy/reference/optimize.html>) to conduct our energy minimization. This method of optimization does not use gradients and cannot incorporate a dissipative energy term. We chose this method due to its suitability for our specific static simulations; this minimization method is able to take large steps in the energy landscape to converge faster and can often recover from divergent energy configurations. To include a dissipative energy term like friction,

we would have to move to a dynamic simulation and use a different optimization scheme, such as gradient descent.

Prior research on rib fabric made of incompressible yarn shows that energy lost to friction is very small, at most the totalling the energy of the third-largest contribution for the entirety of the knit’s elastic response (Duhovic & Bhattacharyya 2006). Since we are using a compressible yarn with an appreciable energy contribution from yarn compression (see Supplementary Fig. 7) and stretching fabric in the quasi-static regime, we estimate that friction has a similarly small, if not smaller, contribution to our fabric.

We do not see frictional effects on the elasticity of the knit fabrics when we iteratively repeat extension experiments, as seen in Supplementary Fig. 3. This lack of measurable frictional effect on the experimental samples indicates that friction must be a very small contribution. This is supported by the fact that we can wear clothes multiple times without them losing their elasticity. Socks in particular retain their elasticity over multiple wears even though they are constantly being stretched and deformed with every step. If friction had a large role in knit fabric mechanics, socks would become single-use items.”

For the mechanics we are studying, the only energy contributions left are bending and compression. We agree that some of the other energy contributions may become important in other situations (such as fabrics made of incompressible yarn or fabric compression) that are beyond the scope of our paper.

The referee makes a good point about tension, which we did not address thoroughly in the paper. We have added the following text to the SI as Supplementary Note 4.6:

“In the simulation, manufacturing tension is relevant during initialization of the fabric before it is stretched or deformed in any way. Others have simulated the actual knitting process (Duhovic & Bhattacharyya 2006), implemented a shrinking factor that reduces the arc-length of segments of yarn until the fabric settles into a rest-state (Kaldor et al. 2008), or taken a picture of a physical sample and used that geometry as an input of the simulation that is then relaxed to near force-balance (Sperl et al. 2022). Many don’t consider tension at all (Ru et al. 2023). Of these initialization strategies, the method we use is closest to that of Sperl et al. (Sperl et al. 2022); we start with a input geometry inspired by the actual geometry of the stitches within the fabric. We then impose constraints and yarn properties. The length constraint, which fixes the length of yarn per stitch, is how we control how tightly the stitches are manufactured. Once these input properties are imposed, we allow the simulation to find the minimum energy configuration that fulfills these constraints. In Supplementary Figure 9, we show how we can control the tightness of the stitches by changing the length of yarn per stitch.

We also consider how tension may affect the yarn structure by allowing the core radius of the yarn to vary. In this way, we account for how the yarn may change its compressibility under tension without a physical model for that phenomenon, which is currently poorly understood. We also take measurements of the yarn radius in situ to represent changes in yarn radius under tension. Worsted weight yarn, such as the acrylic and cotton yarns used in our sample, often visibly change radius under tension.”

We have also included a figure, now Supplementary Figure 9, to show how changing the length constraint affects how tight the stitches are.

(4) In the xx (course) direction in Figure 2, except for plain knitting, the experiments and calculations are different, but I am unable to understand the cause of this difference. For example, if you experiment with a monofilament knitted fabric with high compression stiffness, and/or with a structure where the contact points are fixed, I think the reason will become clearer.

The simulation has six inputs related to the yarn and fabric properties: yarn radius, length of yarn per stitch, the power law of the compression model, the scaling constant for the compression model, the bending modulus, and the core radius of the core-shell compression model. The first two of these inputs are experimentally measured (Supplementary Tables 2 and 3). The following three inputs, the compression model parameters and bending modulus, are derived from fits to experimental data (Supplementary Notes 2 & 3 and Supplementary Table 4). The core radius is a fitting parameter. All of the inputs that are experimentally measured or derived have an associated error. Precisely how this error affects the stress-strain output of the simulation is currently unknown.

In varying the core radius, we find that changing the simulation parameters can affect one direction of

the stress-strain results more than the other direction. It is entirely possible that the errors in each of the simulation inputs has compounded to results in generally softer behavior in the x-direction for the acrylic yarn. If we compare the simulation to experiment for the cotton fabric (Supplementary Fig. 2) in the x-direction, the simulation is generally stiffer than the experiments. This changing relative stiffness between simulation and experiment indicates that it may not be a systemic issue with the simulation, but instead a result of the accuracy of the inputs. We agree that additional research into this problem is necessary, but it is currently beyond the scope of this paper. A systematic study of how error in each simulation input affects the resulting mechanics (alternatively, how yarn and fabric manufacturing properties affect mechanics) is currently underway. You can see an example of preliminary work in Figure 1.

(5) Additional comments

About yarn compression stiffness measurement; Compression is performed using a plane compressor. I believe that the contact during knitting due to yarn contact will change from point contact to area contact. It is unclear whether the same modulus can be used in the model in this paper. About yarn bending stiffness measurement; The bending stiffness of yarn was measured by a cantilever method. I think the curvature of the cantilever is too small compared to the curvature of the yarn in knitted fabrics. Please state why the measurements are valid.

While we agree with the reviewer that there is some inaccuracy with these measurement methods, it is in-line with the constraints of our modeling approach and, we believe, doesn't have a significant impact on our claims.

The compression model that we introduce is phenomenological in that it seeks to capture the effects of changing compression *and* changing contact area. It approximates contact between yarn segments as a collection of point-like contacts. While this approximation is likely a source of some error, it enables us to model a wide range of contact interactions (with changing area, as mentioned), which, given the complexity of such interactions, is difficult to (i) simulate and (ii) characterize to full generality. Our approach allows such an approximation to the full yarn-interaction behavior to be made via a single point compression test. Importantly, since accurate compression behavior is further complicated by differences in yarn microstructure (e.g. spun vs monofilament), and we seek a *material-agnostic* model of fabric mechanics, we opted for a model that combines aspects of the complexity of yarn compression, namely its nonlinear form and changing contact area, with simplicity, i.e. a single modulus that controls the scale of the nonlinear response.

The bending stiffness measurement is similarly used as a simple measure that allows direct comparison of yarn types. There are different bending responses of spun vs monofilament (etc) yarn to applied forces, which would necessitate a broad variety of different yarn models, complete with non-linear, curvature-dependent bending stiffnesses. Approximation of these various yarn types by a single model of bending allows for "apples-to-apples" comparison of bending response, further advancing our goal of a material-agnostic model. Nevertheless, the use of the cantilever method for measuring bending stiffness and applying that measurement to yarns within fabrics has been used by other authors (Cornelissen & Akkerman 2009, Ding et al. 2023, Daelemans et al. 2021), and we find that this method is the most consistent and reproducible way to extract the bending rigidity. As detailed in the Materials and Fabrication section as well as Supplementary Note 2, we measure the bending modulus for different lengths of yarn to cover a wide range of bending. Generally, we have good agreement of the bending modulus regardless of the yarn length and degree of bending. Other methods to extract the bending stiffness exist and have been explored, but pose their own issues. Choi & Lo (2006) use the Kawabata Evaluation System Bending Tester to extract the bending rigidity of their yarn. Although this system bends materials to 150 degrees, it is intended for sheets of fabrics. Alshukar & Macintyre (2020) discuss the inaccuracies of using this machine for yarns and show that the bending rigidity results are often lower than results extracted via the ring-loop test and the beam test. Dhingra (1974) discusses several methods to extract bending stiffness and details the advantages of the cantilever method over the ring-loop method. One disadvantage mentioned of using the cantilever experiments on twisted yarn is yarn untwisting at the free end. In our cantilever tests, we choose yarn lengths that are large enough that we avoid this untwisting. Additionally, the yarns we use are balanced so there is no net torque after the plying process.

Development of our simulation method to achieve greater quantitative fidelity with experimental data ultimately requires further exploration of yarn-specific constitutive modeling. To the degree of combined

generality and accuracy at which we seek to model topology-dependent, rather than material-dependent response, our characterization of compression and bending seems to be sufficient.

Referee #3

The presented manuscript is interesting from the point of view of theoretical modeling, but the practical application is unconvincing. The authors present many examples of functional textiles in the introduction, but the work presented by them has little connection with the cited articles. The introduction must be strengthened by providing a detailed analysis of the knitted structure models that have been developed. It is also necessary to discuss scientific works examining the mechanical behavior of knitted fabrics. There are many developed theoretical models of the knitted loop that allow accurate estimation of the length and shape of the loop, which allows to predict the properties and mechanical behavior of the knitted fabric. List of references and the introductory text must be very close related to the investigated topic.

We acknowledge that we can further discuss fabric modeling in the introduction, and we have added:

“Textiles research has traditionally been housed in both textile engineering and computer graphics; however, the growing interest of textiles as metamaterials in other fields creates the need for cross-disciplinary pollination.”

and

“In this work, our goal is to study knit fabrics from three different types of models: a minimal model of yarn-level simulation at the microscopic level, a constitutive model at the textile level, and our “Reduced-Symmetry” model at the intermediate level to unite these two points of view.”

We have additionally re-written the introduction to the simulation section to better situate our approach and included the following:

“Current simulations typically have at least one of three primary limiting constraints: they do not consider compressible yarn (Abghary et al. 2016, Poincloux et al. 2018, Duhovic & Bhattacharyya 2006), they only consider one fabric type (Abghary et al. 2016, Duhovic & Batthattacharyya 2006, Htoo et al. 2017, Abel et al. 2012, Poincloux et al. 2018), or they only compare simulation to experimental results for visual fidelity and not mechanical response (Ru et al. 2023, Kaldor et al. 2008). Our simulation method considers all three of these factors to investigate the role of stitch topology on the mechanical behavior of knit fabrics.”

However, we also think it is important to well-motivate research in knit mechanics, especially for the broad readership of Nature Communications. The existing introduction is meant to motivate this research

and identify the complex network of fields that are working on knit materials. It is similarly important that we identify the novelty of our work, which uses stitch topology to predict mechanics.

When describing the object of research, the authors must use standardized names of knitted patterns (in this case, they provided only one correct name - rib) and correct expressions. For example, instead of “knitted fabric” (when referring to the specific arrangement of the loops in the knitted fabric) it must be used “knitting pattern” or “pattern”; “stitch gauge” must be “stitch density”, “yarn per stitch” - “loop length”, etc.

Thanks for pointing out that not all readers may understand our terminology. Since we are trying to address a broad readership and are more familiar with hand knitting, we have done our best to include as many terms as possible when discussing each topic.

We were unable to find an alternative term for “seed.” Please let us know if there is an industry standard term we should be using.

I would not agree that when modeling the mechanical behavior of knitwear it is correct to exclude the influence of friction at the points of intersection of yarns, and to analyze only the bending and compression forces. Because the friction between the yarns at the points of intersection of the stitches is one of the factors that help to maintain the specific shape of the stitch. Therefore, friction must be considered in the simulations.

We agree that friction must be used in dynamic simulations. In addition to keeping simulated fabrics together, it is an overdamped system and its behavior depends critically on dissipative forces. Our simulations, however, are not dynamic. For each fixed stitch cell size, the yarn within the stitch is allowed to relax to find the minimum energy configuration (mechanical equilibrium). The simulated stitch is not being continuously stretched. Instead, at every single stitch cell dimension, the yarn relaxes into a minimum energy configuration, as described in Supplementary Note 4.1. As shown in Fig. 3a-j, Fig. S8c-d, Fig. S9, and Fig. S10a-b, friction is not needed to keep the stitches together in static simulations. We have periodic boundary conditions and are not simulating a finite piece of fabric and thus do not have to worry about unravelling.

For clarity, we have added to captions that include outputs of the stitch-level simulations:

- Fig. 3: *“The renderings in (a-j) are repeated unit cells of sample stitch-level simulation outputs.”*
- Supplementary Fig. 6: *“These diagrams were created using sample outputs of the stitch-level simulations.”*
- Supplementary Fig. 8: *“These renderings (c,d) were made using the outputs of the seed simulations.”*
- Supplementary Fig. 10: *“These renderings we done using sample outputs of the stitch-level simulations.”*

We also do not see an appreciable effect from friction in the experimental stress-strain results. These experiments are done in the quasi-static regime, where a single extension experiment takes 1-3min, depending on the fabric type. In Supplementary Fig. 3, we can see that there is no change in the mechanical behavior from repeated experiments. If friction had an appreciable affect on the fabric in this regime, there should be a noticeable trend in the elasticity of the fabric as experiments are repeated. That is not to say that friction doesn't play a role in other regimes, but it does not have a measurable affect on our experiments.

Thus including friction into simulations for the purpose of stitch stability or accurately representing experiments is not supported since we are able to maintain stitch fidelity without it and we don't see friction effects in experiments.

To address concerns about friction, we have added the following text as Supplementary Note 4.7:

“Friction is often included in dynamic simulations of knitted fabrics (Kaldor et al. 2008, Cirio et al. 2017, Duhovic & Bhattacharyya 2006). We do not include friction in our stitch-level simulations, primarily because our simulations are static. Friction can not be incorporated into an energy minimization scheme. Including a dissipative term such as friction into a static simulation is not supported by both the general simulation method and the specific way we minimize the energy.”

For each set of stitch cell dimensions, we iteratively change the shape of the stitch to find the minimum energy configuration (Sokolnikoff 1956), as previously described. We are not continuously stretching the stitch cell. Each set of stitch cell dimensions is a single simulation, unconnected to other simulations of different stitch cell dimensions. Static simulations of this kind well suit our purposes to use simulations to investigate the role of topology on fabric mechanics. By finding the mechanical equilibrium point of the stitch cell for each set of given dimensions, we well represent the mechanics of our experiments. Our stress-strain simulation results are able to replicate the shape of both the linear and non-linear elastic responses of knit fabrics, which has yet to be achieved for knits made of compressible yarns.

As mentioned in Supplementary Note 4.1, we use the Sequential Least Squares Programming (SLSQP) method in the `scipy.optimize` Python package (<https://docs.scipy.org/doc/scipy/reference/optimize.html>) to conduct our energy minimization. This method of optimization does not use gradients and cannot incorporate a dissipative energy term. We chose this method due to its suitability for our specific static simulations; this minimization method is able to take large steps in the energy landscape to converge faster and can often recover from divergent energy configurations. To include a dissipative energy term like friction, we would have to move to a dynamic simulation and use a different optimization scheme, such as gradient descent.

Prior research on rib fabric made of incompressible yarn shows that energy lost to friction is very small, at most the totalling the energy of the third-largest contribution for the entirety of the knit’s elastic response (Duhovic & Bhattacharyya 2006). Since we are using a compressible yarn with an appreciable energy contribution from yarn compression (see Supplementary Figure 7) and stretching fabric in the quasi-static regime, we estimate that friction has a similarly small, if not smaller, contribution to our fabric.

We do not see frictional effects on the elasticity of the knit fabrics when we iteratively repeat extension experiments, as seen in Supplementary Figure 3. This lack of measurable frictional effect on the experimental samples indicates that friction must be a very small contribution. This is supported by the fact that we can wear clothes multiple times without them losing their elasticity. Socks in particular retain their elasticity over multiple wears even though they are constantly being stretched and deformed with every step. If friction had a large role in knit fabric mechanics, socks would become single-use items.”

The authors mentioned that they “we use the elastica model”, however, elastica model can give reasonable accuracy only when using woolen yarns.

Generally, elastica describes nonlinear slender-body theory of curves. In an elastica model of knit stitches, the constituent yarn is continuous and there is an energy cost to bending. The elastica model is thus only dependent on the geometry of the yarn within the stitch and a bending modulus. Other theoretical methods of considering yarn within the stitch include Euler-Bernoulli beam theory (the material is made from homogenous elastic continuum and the bending is tied to the radius of the material), Timoshenko Beam Theory (internal shear also contributes to flexural rigidity), or bead-spring models (the bending energy is dependent on the angle between ‘springs’).

For clarity, we have added the following text to Supplementary Note 4.1:

“Considering knit stitches as elastica – a continuous curve with bending energy – is a well-established method to consider knit fabric geometry (Postle & Munden 1967, Semnani et al. 2003, Kaldor et al. 2008, Ramgulam 2011, Abel et al. 2012). Elastica methods are the middle ground between full three-dimensional continuum elastic models (FEA of the yarn itself) (Kyosev 2012, Abghary et al. 2020) and simplified bead-spring models (Htoo et al. 2017, Sha et al. 2021), originally designed for molecular dynamics of polymers and a method that imposes a non-realistic contact geometry between clasped yarns.”

The Reduced-Symmetry (RS) model we develop in the paper is an elastica model. Because the elastica model is only geometry dependent, it can be used on any inelastic yarn type if the geometry of the rest-state of the stitch is known and we can characterize the bending energy of the yarn. The method we use to measure the bending modulus, the cantilever method, can be used on any type of yarn (Cornelissen & Akkerman 2009). Neither of the two yarn types in our paper are made of wool, yet we are able to characterize their bending (cantilever test) and rest geometry (stitch-level simulations).

Other elastica models of knits are not confined to wool yarn. For example:

- R. B. Ramgulam, Modeling of knitting, in Advances in knitting technology, 48-85, 2011, Woodhead

Publishing

- Postle, R. & Munden, D. L. 24—ANALYSIS OF THE DRY-RELAXED KNITTED-LOOP CONFIGURATION: PART I: TWO-DIMENSIONAL ANALYSIS. *Journal of the Textile Institute* 58, 329–351 (1967).
- Semnani, D., Latifi, M., Hamzeh, S. & Jeddi, A. A. A., A New Aspect of Geometrical and Physical Principles Applicable to the Estimation of Textile Structures: An Ideal Model for the Plain-knitted Loop. *Journal of the Textile Institute* 94, 202–211 (2003).
- Abel, J., Luntz, J. Brei, D. A two-dimensional analytical model and experimental validation of garter stitch knitted shape memory alloy actuator architecture. *Smart Mater. Struct.* 21, 085011 (2012).

The last citation in particular uses elastica theory on shape memory alloy wire, a monofilament, which has a very different structure from a wool yarn.

For the production of the prototype, the authors chose hand knitting. The authors claim that "The knitting machine is ideal for ensuring uniform tension throughout the sample; however, it comes at the expense of not guaranteeing uniform stitch size between types of fabrics. Hand knitting, in contrast, cannot guarantee uniform tension throughout the sample, but provides greater control of stitch size even while altering the pattern of the knit and purl stitches". But this is an incorrect statement, since subjective factors are at work during manual knitting, which in no way guarantee the same length of the knitted loops. Even more, the principle of the glove prototype knitting is very unusual, it is not clear why the authors did not choose a traditional seamless knitting method.

Thank you for identifying that we forgot to cite Supplementary Table 2. We have added the citation to the Supplementary Materials line that you described. As you can see from Supplementary Table 2, the hand-knit samples have a relatively consistent length of yarn per stitch over the four fabric types, ranging from 17.85 ± 0.95 mm (stockinette) to 18.33 ± 1.15 mm (seed). In contrast, the machine knit samples have a much wider variation in the length of yarn per stitch, ranging from 11.28 ± 0.62 mm (stockinette) to 16.10 ± 1.58 mm (rib).

For the new lace-weight samples made on the STOLL industrial knitting machine, we see a similar disparity in the length of yarn per stitch across the four fabric types. To reduce this manufacturing variation, we used different machine stitch size settings for different fabric types (size 12 for stockinette and garter and size 11 for rib and seed). Even with these changes in set stitch size as determined by the knitting machine, the lengths of yarn per stitch are 10-20% longer for rib and seed (in comparison to stockinette) over all five lace-weight yarn types used (Supplementary Table 6).

We have added the following text to the Supplementary Materials:

"For the lace-weight samples made with the STOLL Industrial knitting machine, stockinette and garter were made with a stitch size setting of 12 while rib and seed were made at size 11. We find that if all four types of fabric are made at size 12, rib and seed are significantly more loose (Supplementary Fig. 14)."

In order to maximize the stiffness of the fabric around the part of the wrist we wanted to support with the glove prototype, we desired a fabric orientation parallel to that found in the traditional seamless knitting method. The most rigid fabric and orientation is stockinette in the y-direction (see Supplementary Fig. 10 and Supplementary Table 20). In the traditional seamless knitting method, we could only achieve a maximum fabric rigidity of $\approx 64\%$ of our prototype glove rigidity in the region around the wrist. Knitting the glove as a single flat piece allowed for us to fully utilize the results described in this paper and access the most relevant range of rigidities offered by the four different fabric types. In this way, we have proposed a unique design for a glove that utilizes elastic properties that traditional gloves do not offer. Additionally, we only have access to flat-bed knitting machines. As future work, we need to be able to manufacture this pattern on our flat-bed machines to test variations on this pattern.

Authors present that "We targeted 20 - 30 mm Hg (compression) for the hand of the specific wearer (...). We have achieved this comparable pressure without the use of elastane (!)." It is interesting, how did they achieve this level of compression (this part of the glove must be and how did they measure it? This part of

the glove must be correspondingly smaller in circumference than the wrist of the hand; without elastomeric yarn, such a product is generally difficult to apply to the limb.

We have added the following text to Supplementary Note 10 to clarify how this pressure estimation was done:

“This pressure was calculated by measuring the rest, flat position of the circumference of the glove, then measuring the circumference of the glove on the the hand. From these measurements, we calculate the linear strain of the glove as it is being worn. Using Supplementary Fig. 12, we use this strain to find a correlated stress. Multiplying the stress by the width of the wrist support segment (the stockinette region around the wrist) gives a force, which is then divided by the area of the wrist support segment to estimate a pressure.”

The negative ease of the glove (the portion around the wrist, directly below the thumb joint) is approximately 0.25 in. The glove is not significantly harder to apply to the limb than compression stockings or elastomeric knee braces.

In order to improve the value of the submitted work, I recommend consulting with a textile (knitting) engineering specialist, and also familiarizing yourself with published works that examine real models of knitted loop and properties of knitted structures.

We thank the referee for their suggestion and appreciate their feedback in making the work as accessible as possible. We feel that our work represent an important step in bridging material-specific models of knitting and establishing generic heuristics and mechanical rationale for the role of stitch topology in determining response, independent of fine-tuned yarn material properties and constitutive relations that are often required for more realistic models. Throughout the process, we have been in contact with textile engineering specialists.

Referee #4

This paper explores the relationship between knitted fabric stitch topology and resulting knot elasticity through experiments and simulation. The authors aim at constructing a constitutive model for the nonlinear bulk response of such fabrics. The goal is to untangle the relationship between stitch pattern and mechanical response for the 4 most common knitted fabrics: stockinette, garter, rib and seed. The authors measured the elastic response of each of the four fabrics in a series of uniaxial stretching experiments and simulations, using different types of yarn with different mechanical properties. The simulations show that in the low/high stress regime, bending energy/compression energy is the dominant contributor to elastic response and in the high stress regime. By breaking the problem into a concatenation of two types of stitches it is shown that odd symmetry connecting yarn segments can be of order ten times softer than compared to even symmetry connecting yarn segments.

The presentation of the paper is overall very good and the results are innovative and significant towards understanding of the relation between topology of filaments and mechanics of filamentous matter in general. Therefore, I recommend this paper for publication.

Some more detailed comments are below:

In the caption of Fig.2 it is said that the experimental data are shown in the translucent regions. These are not clear in the figure. The data would be more visible if either the figures are larger or if the lines are thicker.

The translucent regions in Figure 2 show one standard deviation of the experimental data. The small width of these regions represents the precision of the experimental data. The solid lines, by contrast, are fits from the constitutive model. Increasing the thickness of the lines would obscure the experimental data.

In the same also figure, dashed lines are mentioned in the caption, but they are not visible in the figure. Thank you for pointing this out, we have removed the dashed lines reference in the figure caption.

The caption refers to the patterns by their knit names, but it would be helpful if in parenthesis there was a mention of the symbol used in the figure for each of the knits. The figure itself shows the knits, but not the names and the reader has to look back and forth in the paper to make sense of the caption.

We have added the pattern names under the symbols within Figures 2, S2, and S12. We hope this adds clarity.

Finally, in the main text it is said that Fig.2 shows results for different types of yarn. I think it is meant as an average. A few words here about the characteristics of different type of yarn as they are shown in the Supplementary information would be useful.

We are unsure which line the referee was referring to; perhaps the following: “We fabricated and characterized samples made from two types of yarn, an acrylic yarn (Fig. 2) and a pearlized-cotton (Supplementary Fig. 2), which have different mechanical properties (see Methods)”.

We see how there could be confusion here, since the figures have the same number but one figure is in the main text and the other is in the Supplementary Materials. The data in Figure 2a-b is not an average over different yarn types. We have added an additional subplot, Fig. 2c, which displays the average normalized rigidity of each fabric type over samples made of different yarns. The colored ellipses are centered over the average normalized rigidity for each fabric type and the width of the ellipses represents the standard deviation in the data.

REVIEWER COMMENTS

Reviewer #1 (Remarks to the Author):

1. For the stitch geometric model, authors said that "The geometric structure is incorporated within our RS model and yarn-level simulations, but is not directly incorporated into the FEA.". I have doubts about this. Models in the FEA are defined according to the real stitches using coordinates or other data. I know that the geometric shape is not inputs or directly considered by the FEA, but it is the basic of the input data into the FEA. The generation of stitch in Fig.3, Supplementary Fig.6, Supplementary Fig.8 and Supplementary Fig.10 also need these shape data to support the results. Authors also declare that they are working on a second paper to characterize this dependence, it's worthy of recognition. However, it is an independent paper for this manuscript, so I think some brief explanations should be made in this article.

2. Authors cited the references recommended by the reviewers, but only provided a brief description without further explanation of the novelty of this article.

3. Besides gloves, can the author provide several common models using traditional knitting methods for mechanical performance testing?

Reviewer #2 (Remarks to the Author):

In light of current textile research and simulation technology, I believe that my concerns have been reasonably answered and appropriate modifications have been made.

Reviewer #3 (Remarks to the Author):

The manuscript was significantly improved according to suggestions of reviewers and it led to better quality of the presented content. It is extremely important to publish accurate information based on textile engineering knowledge. Misleading or inaccurate information influences appearance of new inaccurate conclusions.

Only one doubt still remains concerning an influence of friction force. Comparison of experimental and simulated stress-strain results presented in Supplementary Figure 2 demonstrate that the highest mismatch of results is in the first part of stress-strain curves, in which stress acts on the changes of the shape of stitch dependent on the movement of intermeshing points between the neighboring stitches (where friction plays a significant role).

Reviewer #5 (Remarks to the Author):

The authors did a fantastic job of responding to the reviewer comments.

You state

"we are not concerned with visual fidelity and large-scale yarn-level simulations. We acknowledge that there is a large amount of literature in computer science that builds up different ad-hoc simulations as a way to model large scale behavior of fabrics. The conclusions drawn in this paper are targeted towards a fundamental understanding of universality between the different stitch types and overall fabric behavior, irrespective of yarn choice. Overall, we feel that we are

answering an orthogonal set of questions than those posed by the computing and engineering community."

I disagree with the last statement about the orthogonality of the technical questions being addressed by the two communities. Things that are completely unrelated are orthogonal. As stated in my first review some researchers in the graphics/CAD/engineering community are concerned with physical fidelity, not just visual fidelity. I think the Z. Liu 2021 paper is a good example of this. They are attempting to model and design the elasticity of a fabric, not its visual appearance.

From my first review.

"I will point out that most of the papers are not about just making pictures or animations. CAD/CAM (Computer-Aided Design/Computer-Aided Manufacturing) has always been a major focus of graphics research. CAD/CAM was the original application of computer graphics.

Since a growing portion of the research from the graphics community is focusing on modeling, simulation and design for making/manufacturing, the methods developed here are concerned about accuracy and physical fidelity, and therefore I believe they should be considered when evaluating a scientific paper about modeling and simulating physical phenomena."

For example the Liu, Han, et al. 2021 paper is interested in your stated goal, understanding how to control the mechanics of a knitted fabric by changing its low-level constituent parts.

I disagree with how you have characterized the Liu, Han 2021 paper. I think a more accurate way to portray it is

"The computer graphics community has made great strides in creating knit fabric simulations with visual fidelity (Kaldor et al. 2008, Kaldor et al. 2010, Cirio et al. 2017), often with the goal of modeling entire sheets (Sperl et al. 2022) of fabric. Recent work in this area (Liu et al. 2021) has also studied how to model and control the elasticity distribution in manufactured knitted garments by changing yarn types in specific regions of the fabric."

You have the following new text in SN 9.

"There is considerable prior work on numerical homogenization of yarn level simulations that use micromechanical simulations to predict the bulk level elastic response that is then implemented in FEA [26, 27]."

Reference 27 should be replaced with

Wadekar P, Perumal V, Dion G, et al. An optimized yarn level geometric model for Finite Element Analysis of weft knitted fabrics. *Comput Aided Geom Des* 2020; 80: 101883.

and

Liu, D., Koric, S. & Kontsos, A. A Multiscale Homogenization Approach for Architected Knitted Textiles. *J. Appl. Mech.* 86, 111006, DOI: 10.1115/1.4044014 (2019).

The work in [27] is not related to FEA.

I strongly believe that

L. Kapllani, C. Amanatides, G. Dion, V. Shapiro and D.E. Breen, "TopoKnit : A Process-Oriented

Representation for Modeling the Topology of Yarns in Weft-Knitted Textiles," Graphical Models, Vol. 118, Paper 101114, October 2021.

should be cited in the section where you reference topology and knitted fabrics. As noted in my first review, the term "topology" in this paper is defined within a Computer-Aided Design (CAD) context (a widely used and understood term in CAD!), which is certainly different than the knot topology in your work. But this is a significant knitting-related topology result from CAD that should be cited. (somewhere)

I appreciate your explanation about the linear bending behavior of a yarn, "Even an order of magnitude difference in bending modulus leads to relatively little change in mechanical behavior of the resulting fabric. Thus, larger order terms in the bending energy are likely to have only a small effect on simulation results."
Is this explanation in the paper somewhere? It should be.

I still feel that you have not effectively justified all of your assumptions about the yarn model. You state "the (yarn) stretching energy is so large that we consider the yarn to be inextensible."

But if the energy is large, doesn't it then have a significant influence on the yarn's mechanics?

"The energy is very large, so we ignore it" argument doesn't make sense to me. Please justify this aspect of your model in your paper.

Can you show that different yarn stretching behaviors have negligible effect on the fabric behavior? Or yarn stretching properties are overwhelmed/dominated by the influence of the stitch topology?

There are numerous errors and omissions in your references. Here are the ones I spotted. Please make sure that all of your citation information is complete and accurate.

In the main paper

[22] is missing a journal title and year.

[38] is missing a journal title and year.

[39] is missing a journal title and year.

[40] is missing a journal title and year.

[41] is missing a journal title and year.

[57] is missing a journal title and year.

[58] is missing a journal title and year.

Remove [59], it is a duplicate of [56].

In the SI

[4] is missing a journal title and year.

[8] is missing a journal title and year.

[10] is missing a journal title and year.

[11] is missing a journal title and year.

[14] is missing a year.

[16] is missing a journal title and year.

[18] is missing a journal title and year.

[20] is missing a journal title and year.

[22] is missing a journal title and year.

[20] and [23] are duplicates. Remove one.

[24] is missing a journal title and year.

This is strong work that should be published, once the authors make the edits and corrections that I have noted.

Reviewer #6 (Remarks to the Author):

The authors made considerable efforts to address not only mine but concerns and comments of all the referees. Overall, I am convinced by the authors' replies and satisfied with the subsequent changes brought to the manuscript. I would therefore recommend publication in Nature Communications without further changes.

However, I still have some trouble with how friction is addressed. These concerns are more in the realm of discussion than actual changes to implement in the manuscript. I fully agree that adding friction to the current model is not easy/feasible and outside of the scope, but I tend to disagree with the arguments put forward.

-Repeatability and elasticity (in the sense of no irreversible deformation upon cycling) of the experiments are not proof that there is no significant frictional dissipation. For instance, the mechanical tests in [45, Poincloux et al 2018] show very high repeatability upon multiple loading-unloading cycles, but the significant hysteresis attests to the energy dissipated by friction. So, garments can have, at the same time, inter-yarn friction and sustain many uses without showing degradation (friction in a sock does not mean that you can wear it once). To provide convincing proof that friction dissipation is fully irrelevant in the present study, the authors could show that their experiments have negligible hysteresis upon cyclic stretching.

-Ignoring friction because the experiments are done in a quasi-static regime is also misleading in my opinion. If friction is implemented numerically as a viscous-like term (dissipation proportional to sliding speed at the contacts), common in computer graphics works for instance, then I agree that a quasi-static approach will drive these frictional terms to zero. But actual fabrics pulled quasi-statically show significant frictional dissipation (see again [45, Poincloux et al 2018] for instance). So inter-yarn friction also has a "dry" component (dissipation proportional to the sliding distance) not suppressed by slow loading. This friction may still lead to a significantly different effective Young modulus between loading and unloading.

Reviewer #7 (Remarks to the Author):

There have been significant additions to the content and references in response to the reviews, which have provided further clarification, however I only specifically address the responses to reviewer #7 comments.

More information has been given about the reasons for yarn choice for the prototype, and also the process of evaluating pressure.

It is noted that the claim regarding the model enabling knitwear designers and soft roboticists to fine tune their products' elastic properties has been removed, and a more realistic ambition and context stated for the application of the project. The ambition for the model following further work has been more clearly stated.

The phrase 'implement on-the-fly changes' has been removed in one instance (previous p5 line 147) and replaced with a new explanation, however the phrase is still used on p1, line 27 and should be replaced there as well with a more explanatory phrase

In original Supplementary note 3 (now Note 11) there is no change related to the notion of 'greater control', although a point was made in the rebuttal letter. I would argue that the 'fixed diameter needle' is equally applicable to the knitting machine hooked needles for any specific gauge, and needle size does not completely dictate the hand knitting tension, as the work of different individual knitters' would vary in this respect. Here only one knitter has presumably been used. I would like to see more clarity on these points in Supplementary note 11.

Other more minor points have been addressed, although the depiction in Figure 1 of a single bed knitting machine has not been justified or explained in the text, but rebutted, which I regret.

Therefore following further attention to these final points, the manuscript would be acceptable from this reviewer's perspective.

Responses to Referees

Jan 2024

We thank all the reviewers for their additional feedback on our manuscript. We hope, after careful consideration and revision, that this manuscript remains of interest to Nature Communications. We would like to address the comments of the referees as follows (our comments are in blue):

Changes Made:

- We have added an additional figure, now Supplementary Fig. 12, to clarify the form of the FEA calculations and how the constitutive relations were applied to a large-scale sheet. We have included reference to this new figure in the main text and updated figure references for all succeeding Supplementary Figures.
- We have included additional details on the state of the field in the computer graphics community.
- We fixed an incorrectly cited article and also corrected an error in our referencing system so that references correctly appear with the journal title and year.
- We have clarified why we do not consider yarn extension.
- We clarified that all of our results and analysis are applicable to fabrics being stretched, not un-loaded, and that the stitch-level simulations cannot model fabric un-loading (see additions to Supplementary Note 4.1).
- We removed a previously overlooked instance of “on-the-fly”.
- We have clarified how hand-knitting provides better consistency in length of yarn per stitch over different fabric types.
- We now explicitly state in the caption of Figure 1 that the second needle bed (ribber) is not shown in the inset of Fig. 1a.

Reviewer #1

1. For the stitch geometric model, authors said that “The geometric structure is incorporated within our RS model and yarn-level simulations, but is not directly incorporated into the FEA.”. I have doubts about this. Models in the FEA are defined according to the real stitches using coordinates or other data. I know that the geometric shape is not inputs or directly considered by the FEA, but it is the basic of the input data into the FEA. The generation of stitch in Fig. 3, Supplementary Fig. 6, Supplementary Fig. 8 and Supplementary Fig. 10 also need these shape data to support the results. Authors also declare that they are working on a second paper to characterize this dependence, it’s worthy of recognition. However, it is an independent paper for this manuscript, so I think some brief explanations should be made in this article.

For clarity, we have added a figure, now Supplementary Figure 12, to show the mesh that the FEA calculation is using. Subplot (a) shows the un-deformed mesh, subplot (b) shows the deformed mesh, and subplot (c) shows the trace of the strain tensor squared over the deformed sheet. The figures you cited are generated from results of the stitch-level, Bézier-spline simulations and are not related to the FEA.

2. Authors cited the references recommended by the reviewers, but only provided a brief description without further explanation of the novelty of this article.

We have explicitly identified the novelty of our research (in comparison to the existing body of work) in the following lines:

“Here, inspired by the design of hand-knit garments, we study how the mechanical behavior of weft knitted fabrics is encoded by the topology of their stitches as a first step towards creating a design tool for programmable textile metamaterials.” (lines 34-36) (these lines are preceded by text on the growing field of programmable metamaterials)

“There is some recent work on changing local fabric elasticity by changing the constituent yarn (Liu et al. 2021), but there has not yet been a systematic study of how changes in stitch topology affect the fabric elasticity (Tekerek et al 2020) – even modeling stockinette (sometimes called jersey or plain-knit) fabric is quite complex (Choi & Lo 2006, Postle 2002, Poincloux et al. 2018). In this work, our goal is to study knit fabrics from three different types of models: a minimal model of yarn-level simulation at the microscopic level, a constitutive model at the textile level, and our “Reduced-Symmetry” model at the intermediate level to unite these two points of view.” (lines 41-47)

“Our goal is to use stitch type as a way of modulating the bulk elasticity of fabrics made of inelastic yarn, irregardless of fiber composition, so that the desired elastic response of a textile can be achieved with natural and/or biodegradable fibers and without synthetic materials. Recent research has shown that a broad range of synthetic materials can degrade when in contact with skin secretions, which increases the potential for dermal absorption of compounds within those fibers (Abafe et al. 2023).” (lines 51-56)

“Current simulations typically have at least one of three primary limiting constraints: they do not consider compressible yarn (Abghary et al. 2016, Poincloux et al. 2018, Duhovic & Bhattacharyya 2006), they only consider one fabric type (Abghary et al. 2016, Duhovic & Batthattacharyya 2006, Htoo et al. 2017, Abel et al. 2012, Poincloux et al. 2018), or they only compare simulation to experimental results for visual fidelity and not mechanical response (Ru et al. 2023, Kaldor et al. 2008). Our simulation method considers all three of these factors to investigate the role of stitch topology on the mechanical behavior of knit fabrics.” (lines 97-101)

In these examples, we provide the current state of the field and include how we are furthering the body of research in this particular article. If you were referring to the novelty of the references we included, unfortunately we are constrained by length and cannot give more detailed descriptions of all of our references.

3. Besides gloves, can the author provide several common models using traditional knitting methods for mechanical performance testing?

Vu & Kim 2020 (citation 20 in the main text) utilized knitting to incorporate pressure sensors into both gloves and socks. In this example, the mechanical behavior of knit fabrics is used to collect data.

We used hand-knitting for the glove primarily because, at the time, we did not have access to a programmable machine of an appropriate gauge. It was both faster and easier to develop the pattern through hand-knitting than to use the Taitexma Industrial Knitting Machine (which is not computerized). Now that we have the pattern, we will be able to use a modified version with the STOLL Industrial Knitting Machine, which is computerized but only knits with yarn of a finer weight. That being said, there are situations in which machine knitting isn’t possible; for example, in Magnan et al. 2020 (citation 9 in the main text), they use hand knitting to knit a sheet of fabric from thread made of human skin cells.

Reviewer #2

In light of current textile research and simulation technology, I believe that my concerns have been reasonably answered and appropriate modifications have been made.

Thank you!

Reviewer #3

The manuscript was significantly improved according to suggestions of reviewers and it led to better quality of the presented content. It is extremely important to publish accurate information based on textile engineering knowledge. Misleading or inaccurate information influences appearance of new inaccurate conclusions.

Only one doubt still remains concerning an influence of friction force. Comparison of experimental and simulated stress-strain results presented in Supplementary Figure 2 demonstrate that the highest mismatch of results is in the first part of stress-strain curves, in which stress acts on the changes of the shape of stitch dependent on the movement of intermeshing points between the neighboring stitches (where friction plays a significant role).

In the low-strain discrepancies you identify in Supplementary Figure 2, the simulation results are typically stiffer than the experiment. If these discrepancies were due to friction, we would expect that the experimental data would be stiffer than simulation, since friction would act against the fabric stretching and lead to a stiffer elastic response. Instead, we think that geometric confinement leads to this low-stress discrepancy, as described in Supplementary Note 4.4.

We implemented a version of static friction in the simulations for seed fabric, where we constrained the arc-length to prevent yarn sliding. This implementation is described in Supplementary Note 4.2. This static friction did not inhibit the simulation from geometric confinement since we still see low-stress-high-modulus behavior in the stress-strain results for seed in Supplementary Figure 2.

We have added the following text to Supplementary Note 4.1:

“While static friction is expected to play an important role in various aspects of fabric mechanics, our focus on using energy-minimizing configurations of stitches to approximate both the “relaxed” ($f = 0$) and “deformed” ($f \neq 0$) states of the stitches led us to develop simulations that are inherently different than the use of simulated damping from friction (i.e. gradient descent) to “evolve” configurations towards equilibrium states. This simplification allows us to focus on details of yarn geometry in determining equilibrium response, without needing to worry about specifying certain deformation paths. Given that the Bézier representation uses control points that provide non-local control over yarn shape, it is difficult to incorporate damping terms in these simulations; representations that yield local control, such as other B-splines, would be more suited for investigating friction and path-dependence of deformation protocols (e.g. loading vs. unloading). However, we do find that some level of static friction is necessary to reproduce the deformation of seed stitches; we introduce constraints in order to simulate this effect (see next section).”

Reviewer #5

The authors did a fantastic job of responding to the reviewer comments.

You state

“we are not concerned with visual fidelity and large-scale yarn-level simulations. We acknowledge that there is a large amount of literature in computer science that builds up different ad-hoc simulations as a way to model large scale behavior of fabrics. The conclusions drawn in this paper are targeted towards a fundamental understanding of universality between the different stitch types and overall fabric behavior, irrespective of yarn choice. Overall, we feel that we are answering an orthogonal set of questions than those posited by the computing and engineering community.”

I disagree with the last statement about the orthogonality of the technical questions being addressed by the two communities. Things that are completely unrelated are orthogonal. As stated in my first review some researchers in the graphics/CAD/engineering community are concerned with physical fidelity, not just

visual fidelity. I think the Z. Liu 2021 paper is a good example of this. They are attempting to model and design the elasticity of a fabric, not its visual appearance.

From my first review. “I will point out that most of the papers are not about just making pictures or animations. CAD/CAM (Computer-Aided Design/Computer-Aided Manufacturing) has always been a major focus of graphics research. CAD/CAM was the original application of computer graphics.

Since a growing portion of the research from the graphics community is focusing on modeling, simulation and design for making/manufacturing, the methods developed here are concerned about accuracy and physical fidelity, and therefore I believe they should be considered when evaluating a scientific paper about modeling and simulating physical phenomena.”

For example the Liu, Han, et al. 2021 paper is interested in your stated goal, understanding how to control the mechanics of a knitted fabric by changing its low-level constituent parts.

I disagree with how you have characterized the Liu, Han 2021 paper. I think a more accurate way to portray it is

“The computer graphics community has made great strides in creating knit fabric simulations with visual fidelity (Kaldor et al. 2008, Kaldor et al. 2010, Cirio et al. 2017), often with the goal of modeling entire sheets (Sperl et al. 2022) of fabric. Recent work in this area (Liu et al. 2021) has also studied how to model and control the elasticity distribution in manufactured knitted garments by changing yarn types in specific regions of the fabric.”

To address your concerns on how we address Liu et al. 2021, we have added the following bolded section to the main text:

“There is recent work on changing local fabric elasticity by changing the constituent yarn (Liu 2021), but there has not yet been a systematic study of how changes in stitch topology affect the fabric elasticity (Tekerek 2020) – even modeling stockinette (sometimes called jersey or plain-knit) fabric is quite complex (Choi 2006, Postle 2002, Poincloux 2018).”

You have the following new text in SN 9. “There is considerable prior work on numerical homogenization of yarn level simulations that use micromechanical simulations to predict the bulk level elastic response that is then implemented in FEA [26, 27].”

Reference 27 should be replaced with

Wadekar P, Perumal V, Dion G, et al. An optimized yarn level geometric model for Finite Element Analysis of weft knitted fabrics. *Comput Aided Geom Des* 2020; 80: 101883.

and

Liu, D., Koric, S. & Kontsos, A. A Multiscale Homogenization Approach for Architected Knitted Textiles. *J. Appl. Mech.* 86, 111006, DOI: 10.1115/1.4044014 (2019).

The work in [27] is not related to FEA.

Thank you for pointing out that we cited the wrong Wadekar et al. 2020 paper. We have corrected this error, and also included the Liu et al. 2019 citation at this location.

I strongly believe that

L. Kapllani, C. Amanatides, G. Dion, V. Shapiro and D.E. Breen, “TopoKnit : A Process-Oriented Representation for Modeling the Topology of Yarns in Weft-Knitted Textiles,” *Graphical Models*, Vol. 118, Paper 101114, October 2021.

should be cited in the section where you reference topology and knitted fabrics. As noted in my first review, the term “topology” in this paper is defined within a Computer-Aided Design (CAD) context (a widely used and understood term in CAD!), which is certainly different than the knot topology in your work. But this is a significant knitting-related topology result from CAD that should be cited. (somewhere)

Respectfully, we feel that discussing an alternative notion of topology would be confusing to the reader-ship.

I appreciate your explanation about the linear bending behavior of a yarn, “Even an order of magnitude difference in bending modulus leads to relatively little change in mechanical behavior of the resulting fabric.

Thus, larger order terms in the bending energy are likely to have only a small effect on simulation results.” Is this explanation in the paper somewhere? It should be.

This is unpublished data from a different project and it will not be included in this paper. Without this data, we could not make this claim so we will not be including it in the paper.

I still feel that you have not effectively justified all of your assumptions about the yarn model. You state “the (yarn) stretching energy is so large that we consider the yarn to be inextensible.” But if the energy is large, doesn’t it then have a significant influence on the yarn’s mechanics? “The energy is very large, so we ignore it” argument doesn’t make sense to me. Please justify this aspect of your model in your paper. Can you show that different yarn stretching behaviors have negligible effect on the fabric behavior? Or yarn stretching properties are overwhelmed/dominated by the influence of the stitch topology?

For clarity, we have removed that sentence and added the following bold text to the main paper:

“The torsional rigidity of a balanced, spun yarn is comparatively negligible, **so our model allows the yarn to freely twist. Similarly, the extensional rigidity of the yarn is taken to be large so that the stretch of the yarn plays a minimal role in the fabric’s ability to stretch.**”

For an example of the extensional rigidity, Ding et al. 2023 (<https://arxiv.org/abs/2307.12360>) reports an extensional rigidity of $\sim 80 \text{ N mm}^{-1}$ for an acrylic spun yarn. This is 1-2 orders of magnitude larger than the rigidities of the fabrics reported in our paper.

There are numerous errors and omissions in your references. Here are the ones I spotted. Please make sure that all of your citation information is complete and accurate.

In the main paper

[22] is missing a journal title and year.

[38] is missing a journal title and year.

[39] is missing a journal title and year.

[40] is missing a journal title and year.

[41] is missing a journal title and year.

[57] is missing a journal title and year.

[58] is missing a journal title and year.

Remove [59], it is a duplicate of [56].

In the SI

[4] is missing a journal title and year.

[8] is missing a journal title and year.

[10] is missing a journal title and year.

[11] is missing a journal title and year.

[14] is missing a year.

[16] is missing a journal title and year.

[18] is missing a journal title and year.

[20] is missing a journal title and year.

[22] is missing a journal title and year.

[20] and [23] are duplicates. Remove one.

[24] is missing a journal title and year.

Thank you for pointing this out. We have found an error in our referencing system that has been corrected, and all of the journal titles and years have now made it into the references list.

This is strong work that should be published, once the authors make the edits and corrections that I have noted.

Reviewer #6

The authors made considerable efforts to address not only mine but concerns and comments of all the referees. Overall, I am convinced by the authors' replies and satisfied with the subsequent changes brought to the manuscript. I would therefore recommend publication in Nature Communications without further changes. However, I still have some trouble with how friction is addressed. These concerns are more in the realm of discussion than actual changes to implement in the manuscript. I fully agree that adding friction to the current model is not easy/feasible and outside of the scope, but I tend to disagree with the arguments put forward.

-Repeatability and elasticity (in the sense of no irreversible deformation upon cycling) of the experiments are not proof that there is no significant frictional dissipation. For instance, the mechanical tests in [45, Poincloux et al 2018] show very high repeatability upon multiple loading-unloading cycles, but the significant hysteresis attests to the energy dissipated by friction. So, garments can have, at the same time, inter-yarn friction and sustain many uses without showing degradation (friction in a sock does not mean that you can wear it once). To provide convincing proof that friction dissipation is fully irrelevant in the present study, the authors could show that their experiments have negligible hysteresis upon cyclic stretching.

-Ignoring friction because the experiments are done in a quasi-static regime is also misleading in my opinion. If friction is implemented numerically as a viscous-like term (dissipation proportional to sliding speed at the contacts), common in computer graphics works for instance, then I agree that a quasi-static approach will drive these frictional terms to zero. But actual fabrics pulled quasi-statically show significant frictional dissipation (see again [45, Poincloux et al 2018] for instance). So inter-yarn friction also has a "dry" component (dissipation proportional to the sliding distance) not suppressed by slow loading. This friction may still lead to a significantly different effective Young modulus between loading and unloading.

You have raised valid points about the scope of our simulation and friction argument, which we could more clearly acknowledge within the text. We do not argue that knit fabrics do not experience friction, merely that friction is a small contribution and that yarn bending and compression have a much larger impact on the mechanics of knit fabrics that are uniaxially stretched. Un-loading is a mechanical behavior where friction is quite important. Our simulations cannot model un-loading, as they have no knowledge of previous states of the system. Each set of stitch cell dimensions is simulated independent from all the others. In order to model un-loading, friction must be considered and a different simulation method would have to be used. For this reason, we are only doing analysis on loading conditions, and not analyzing any mechanics of un-loading.

To better clarify the scope of our results to loading conditions and not un-loading conditions, we have added the following to the main text:

"All uniaxial stretching experiments in this work only consider loading; we do not consider the cases of unloading."

We have additionally added to Supplementary Note 4.1:

"While static friction is expected to play an important role in various aspects of fabric mechanics, our focus on using energy-minimizing configurations of stitches to approximate both the "relaxed" ($f = 0$) and "deformed" ($f \neq 0$) states of the stitches led us to develop simulations that are inherently different than the use of simulated damping from friction (i.e. gradient descent) to "evolve" configurations towards equilibrium states. This simplification allows us to focus on details of yarn geometry in determining equilibrium response, without needing to worry about specifying certain deformation paths. Given that the Bézier representation uses control points that provide non-local control over yarn shape, it is difficult to incorporate damping terms in these simulations; representations that yield local control, such as other B-splines, would be more suited for investigating friction and path-dependence of deformation protocols (e.g. loading vs. unloading). However, we do find that some level of static friction is necessary to reproduce the deformation of seed stitches; we introduce constraints in order to simulate this effect (see next section)."

Reviewer #7

There have been significant additions to the content and references in response to the reviews, which have provided further clarification, however I only specifically address the responses to reviewer #7 comments.

More information has been given about the reasons for yarn choice for the prototype, and also the process of evaluating pressure.

It is noted that the claim regarding the model enabling knitwear designers and soft roboticists to fine tune their products' elastic properties has been removed, and a more realistic ambition and context stated for the application of the project. The ambition for the model following further work has been more clearly stated.

The phrase 'implement on-the-fly changes' has been removed in one instance (previous p5 line 147) and replaced with a new explanation, however the phrase is still used on p1, line 27 and should be replaced there as well with a more explanatory phrase

We missed this iteration of 'on-the-fly' when editing and have removed this phrasing in the main text.

In original Supplementary note 3 (now Note 11) there is no change related to the notion of 'greater control', although a point was made in the rebuttal letter. I would argue that the 'fixed diameter needle' is equally applicable to the knitting machine hooked needles for any specific gauge, and needle size does not completely dictate the hand knitting tension, as the work of different individual knitters' would vary in this respect. Here only one knitter has presumably been used. I would like to see more clarity on these points in Supplementary note 11.

We have removed the phrase 'greater control' and replaced it in the text. This sentence in Supplementary Note 11 now reads:

"Hand knitting, in contrast, cannot guarantee uniform tension throughout the sample but **results in more uniform** stitch size even while altering the pattern of the knit and purl stitches due to the fixed diameter of the knitting needle, as seen in Supplementary Table 2."

As discussed in Supplementary Note 11, the knitting machine produces fabric samples with more variable length of yarn per stitch for different fabric types, whereas knitting by hand produces more consistent length of yarn per stitch for different fabric types. We hope the change in wording listed above makes this more clear.

Other more minor points have been addressed, although the depiction in Figure 1 of a single bed knitting machine has not been justified or explained in the text, but rebutted, which I regret.

We have updated the caption for Fig. 1 to include the following:

"Here, the second bed of the knitting machine (the ribber) has been removed for clarity."

Therefore following further attention to these final points, the manuscript would be acceptable from this reviewer's perspective.

REVIEWERS' COMMENTS

Reviewer #3 (Remarks to the Author):

I confirm that my comments have been properly answered and appropriate changes have been made in the presented article. The article is suitable for publication.

Reviewer #3 (Remarks on code availability):

Yes, I was able to install and run the code.

Reviewer #5 (Remarks to the Author):

The authors have adequately addressed my concerns.

I believe that this excellent paper should be published.

Reviewer #6 (Remarks to the Author):

I am fully satisfied with the current version of the manuscript and appreciate the effort of the authors to address my comments. I therefore recommend publication in Nature Communications without further change.